# Mass Start or Time Trial? Structure of the Nervous System and Neuroregeneration in *Pygospio elegans* (Spionidae, Annelida)

**DOI:** 10.3390/biology12111412

**Published:** 2023-11-09

**Authors:** Ksenia V. Shunkina, Zinaida I. Starunova, Elena L. Novikova, Viktor V. Starunov

**Affiliations:** 1Zoological Institute RAS, Saint Petersburg 199034, Russia; keyelastic@gmail.com (K.V.S.);; 2Faculty of Biology, St. Petersburg State University, Saint Petersburg 199034, Russia

**Keywords:** Annelida, regeneration, nervous system, confocal microscopy, serotonin, FMRFamide

## Abstract

**Simple Summary:**

We study how a common marine spionid, *Pygospio elegans*, regenerates its nervous system at the anterior and posterior ends. We use immunostaining techniques with antibodies against serotonin and FMRFamide to label the nervous system of intact and regenerating worms. Our findings show that the regeneration of the *P. elegans* central nervous system has common features with that of other annelids, while the regeneration of the peripheral nervous system depends on individual features. Comparing these findings with the results for other annelids provides valuable insights into both conservatism and plasticity in the mechanisms of nervous system regeneration.

**Abstract:**

The spionid worm *Pygospio elegans* is a convenient model for regeneration studies due to its accessibility, high tolerance, and ease of maintenance in laboratory culture. This article presents the findings regarding neuroregeneration and the structure of the nervous system based on antibody labeling of serotonin and FMRFamide. We propose the main stages of central nervous system neurogenesis during regeneration: single nerve fibers, a loop structure, and neurons in the brain and segmental ganglia. Nerve fibers and receptor cells of the peripheral nerve system can be traced to different stages of regeneration. We also provide a comparison of our results with previous data on the structure and regeneration of the nervous system based on antibody labeling of catecholamines, gamma-aminobutyric acid, and histamine and with the results for other annelids.

## 1. Introduction

Spionidae are one of the largest and the most common polychaete families. The ability to reproduce asexually via architomy or paratomy and thus the high regenerative potencies of some spionid species have been the focus of attention [1,2,3,4]. Posttraumatic regeneration in nature occurs when parts of the body are lost or damaged; this is possible for both the anterior and the posterior ends of worms. Regeneration following injury and fragmentation (in the case of asexual reproduction) has been shown to follow similar morphogenetic patterns [3]. This advanced ability to regenerate has now been extensively explored in around 30 spionid species [5,6].

*Pygospio elegans* is a typical spionid, building tubes in shallow waters at the soft bottom of the intertidal shore. *P. elegans* is among the most frequently employed spionid species and has been studied in great detail in various investigations, including those on regeneration [2,7,8,9]. The convenience of using this spionid is due to its accessibility, small size, and ease of maintenance in laboratory culture [10] as well as its tolerance to fluctuations in temperature, salinity, light, and other abiotic factors [11]. All these features make *P. elegans* a convenient model for studying regeneration since experiments can be conducted under controlled laboratory conditions. The results obtained during such experiments can be easily compared with those on other species, especially model organisms. Nowadays, several annelid species can be considered model organisms for developmental and regeneration studies. The best studied species being *Platynereis dumerilii* [12,13,14]. The experimental conditions for it are standardized, facilitating straightforward comparisons and interpretations [15].

The invention of the confocal microscope rejuvenated comparative annelid neuromorphology, making it more precise and selective [16]. Together with immunohistochemical labeling, it allowed researchers to make volume renderings of the thinnest nerves, which contributed a lot to our acquisition of knowledge, especially in the fields of developmental biology and regeneration studies. Compared to classical techniques, such as methylene blue staining [17] or silver impregnation [18], immunohistochemical labeling has the advantage of yielding more reproducible results; another merit being that the antibodies label only the specific substances they were raised against.

In the past two decades, the combination of antibodies against acetylated α-tubulin, serotonin (5-hydroxytryptamine, 5-HT), and the neuropeptide FMRFamide have become the most popular choices in studies of annelid nervous systems [19,20,21,22]. Acetylated α-tubulin is a structural protein that allows visualization of microtubules in neurites. 5-HT is a common monoamine with several functions, but in the nervous system, it usually functions as a neurotransmitter. FMRFamide is a neuropeptide member of a large RFamide-like family. The peptides of this family are responsible for regulating a wide range of physiological processes, including neurotransmission, neuromodulation, and signaling [23,24]. These techniques have proved invaluable in understanding the neuromorphology of annelids and have contributed significantly to developmental biology and regenerative studies [25,26,27].

During anterior and posterior posttraumatic regeneration in *P. elegans*, the following regenerative stages have been described: wound healing, blastema, formation of the head and pygidium primordia, segmentation, differentiation, and growth. These stages are comparable in various annelid species capable of posttraumatic or fragmentation-induced regeneration, though their timing is different [10,28,29,30]. The tempo and duration of regeneration depend on the species, the size of the worm, its stage of development (larva, juvenile, adult), the temperature during experiments, and the ablated parts and the size of these parts [3,4,31,32]. Numerous investigations have been conducted on regeneration of various species of annelids [5,6,33,34], but there is much less research on nervous system regeneration [35]. Most of these studies focus on the general topology of the nervous system by labeling only acetylated α-tubulin [36,37] or by using the most common antibodies against 5-HT and FMRFamide [20,28,30,38,39]. Only a few studies have focused on less common substances of the nervous system, for instance, peptides such as substance P and GLWamide [40], neuropeptide Y, and proctolin [39], catecholamine (CA) [41], or gamma-aminobutyric acid (GABA) [42].

Here we present the results of the study of *P. elegans* neuroregeneration based on antibody labeling of 5-HT and FMRFamide. It is the continuation of our research dedicated to the regeneration of CA- [10], GABA- and histamine (HA)-positive elements [43] in this spionid. In all our experiments, we used the same conditions (temperature, light regime, type and size of ablation, etc.), and the external and internal morphological characteristics corresponded precisely to all regeneration stages; thus, we provide a detailed comparison of our results.

## 2. Materials and Methods

### 2.1. Animal Collection and Keeping

*Pygospio elegans* Claparède, 1863 worms were collected at the mid intertidal zone from mud and sand flats in the Barents Sea region, near the Marine Biological Station at Dalnie Zelentsi (69°07′ N, 36°05′ E). The worms were transported to the laboratory and kept at 8 °C in plastic containers with prepared sieved sand (≥0.5 mm). The artificial seawater (Red Sea Coral Pro salt, Israel) with a salinity of about 30–32‰ was changed once a week. The animals were fed with homogenized spinach once a week after the water change.

### 2.2. Sampling and Regeneration Experiments, Fixation

For a general description of the structure of the *P. elegans* nervous system, 20 worms with approximately 40 (±2) setigers, about 1 mm in size, were selected. For staining, we chose worms with no visible damage and no signs of previous regeneration. For regeneration experiments, worms of the same size and number of segments were selected. The *P. elegans* body can reach 70–80 segments in length. We considered the worms with 20–30 segments to be too young (larvae usually settle at the 18-segment stage). Worms with more than 50 segments were considered too old for the experiment. For regeneration experiments, worms were dissected with razor blades after the 19th body segment (Figure 1). The two parts of the worm were kept in separate Petri dishes filled with artificial seawater at 18 °C. We used the following fixation time points: 0 h post-amputation (hpa), 4 hpa, 12 hpa, 18 hpa, 24 hpa, 48 hpa, 3 days post-amputation (dpa), 4 dpa, and 7 dpa. Additionally, we added extra time points of 5, 6, and 9 dpa. No less than 10 worms were fixed for each type of antibody at each time point of regeneration. All animals were relaxed in a 7.5% solution of MgCl_2_•6H_2_O in distilled water before fixation. Relaxed animals were fixed for 12 h in 4% paraformaldehyde solution in 0.1 M phosphate-buffered saline (PBS) at 4 °C. After fixation, samples were washed for 3 × 20 min in 0.1 M PBS with 0.1% Triton X-100 (PBT) and then kept in 0.1 M PBS containing 0.03% sodium azide at 4 °C.

### 2.3. Immunochemistry

Before staining, the worms were rinsed in PBT and preincubated overnight in PBT with 0.25–1% bovine serum albumin (BSA) at 4 °C. We used the following primary antibodies: polyclonal rabbit anti-serotonin IgG (Immunostar (Hudson, WI, USA), 20080) and polyclonal rabbit anti-FMRFamide IgG (Immunostar, 20091). The primary antibodies were diluted 1:1000–1:2000 in PBS with 0.25% BSA. To label the overall nerve scaffold, the monoclonal mouse anti-acetylated α-tubulin antibody IgG (Sigma (San Francisco, CA, USA), T-6793) was added at a 1:2000 dilution. The specimens were incubated in primary antibodies for 16–24 h at 4 °C, then washed for 3 × 20 min in PBT and incubated overnight at 4 °C with secondary antibody mix containing Alexa Fluor 488 Donkey Anti-Rabbit (Molecular (Eugene, OR, USA), A-21206) and Alexa Fluor 647 Donkey Anti-Mouse probes (Molecular probes, A-31571) both diluted 1:800–1:1000. Then, the specimens were again washed several times in PBT, mounted in Mowiol 4-88 with DAPI (Carl Roth (Karlsruhe, Germany) 6335.1), placed between two coverslips, and then stored at 4 °C before microscopy.

The application of anti-serotonin, anti-FMRFamide, and anti-acetylated α-tubulin antibodies ensured reproducible results. We conducted staining for each time point of regeneration not less than three times, analyzing slides from multiple individuals. Our results showed only elements of the nervous system that exhibited consistency across multiple samples. Notably, we observed no differences in the nervous system’s structure or staining between the intact parts of the regenerated worms and the uninjured specimens.

The antibody specificity was tested in a series of control experiments. The negative controls without primary or secondary antibodies resulted in the absence of a fluorescent signal. In addition, preabsorbtion of anti-5-HT-antibody with 40 µm/mL of a BSA-conjugated serotonin control (Immunostar, 20081) led to the complete elimination of the fluorescent signal.

### 2.4. Microscopy and Image Processing

Specimens were examined using a Leica TCS SP5 laser-scanning microscope (Leica Microsystems, Wetzlar, Germany). The specimens were scanned in 50–70 optical sections with a thickness of 1 µm. The resulting confocal stacks were processed with Fiji [44] or Bitplane IMARIS. When necessary, the brightness and contrast of the resulting images were adjusted with Krita. The schemes were drawn in Inkscape.

## 3. Results

### 3.1. The General Structure of the Nervous System

#### 3.1.1. 5-HT-Positive Immunoreactivity

The somata of 5-HT-positive neurons were found mainly in the central nervous system, both in the brain and the ventral nerve cord of *P. elegans*. The brain possessed up to eight pairs of 5-HT-positive somata (Figure 2B,D): one or two pairs in the anterior part, in front of the ventral commissure; 4 pairs surrounding the central neuropil, and two pairs near the dorsal roots of the circumesophageal connectives. The processes of 5-HT-positive cells formed a dense meshwork in the central neuropil and outlined most of the major head nerves as well as the ventral and the dorsal roots of circumesophageal connectives (Figure 2A–E).

In the palps, the processes of 5-HT-positive neurons contributed to the main palp nerve as well as to the four additional nerves (Figure 2A,F–H). The most prominent was the main palp nerve, to which several 5-HT-positive fibers contributed, while the additional nerves usually possessed only one immunoreactive process each. We were not able to locate the exact origin of these fibers; however, they were traced up to the dorsal commissure (Figure 2A–C). The dorsal nerve of the palp never formed arborizations, while the lateral nerves could branch and anastomose with the neighboring nerves (Figure 2F,H). On the inner side of the palp above the food groove, there were two rows of 5-HT-immunoreactive bipolar cells. Their axons entered the main palp nerve alternately from two sides, like a zipper (Figure 2F,G).

A large number of 5-HT-positive fibers were found in the head nerves (Figure 2C–E). Their arborizations formed a thin meshwork of fibers in the prostomium, which was clearly visible both on the dorsal and the ventral sides. The most prominent were frontal prostomial nerves or stomatogastric nerves in Orrhage’s classification (Figure 2D,E). They ran along the dorsal side of the prostomium and wrapped ventrally to the dorsal wall of the mouth. These nerves were then traced along the lateral sides of the pharynx (Figure 3A,B,D). In the anterior part of the pharynx, these nerves were connected by a loose commissure. In the posterior part of the pharynx, on the border with the midgut, they fused, forming a commissure or a loop on the ventral side of the intestine (Figure 3D,E). Thin nerve fibers emanated from this intestinal commissure, forming a network of 5-HT-positive fibers in the midgut (Figure 3E). 5-HT-positive immunoreactive fibers abundantly innervated the pharynx itself. They formed a loose network surrounding the pharynx. Inside the epithelium of the anterior part of the pharynx, we found numerous 5-HT-positive receptor cells. They were close to a triangular shape with somata located at the base of the layer, and the receptor processes extended to the surface (Figure 3C).

The circumesophageal connectives contained numerous 5-HT-positive fibers (Figure 2E). Within the subesophageal ganglion, we found a notable aggregation of 5-HT-positive neurons alongside two well-defined commissures (Figure 4A). At the same time, 5-HT-positive fibers did not practically contribute to the segmental nerves, with only single fibers being distinguishable (Figure 4B). This pattern persisted in the subsequent segments of the body. There were relatively few 5-HT-positive cell somata in the body segments as well as single fibers that contributed to the commissures (Figure 5A).

There was a relatively dense network of thin 5-HT-positive fibers in the body wall. On the sides of the segments, posterior to the parapodia, dense tangles of 5-HT-positive fibers were found in the glands, located at the parapodia bases (Figure 4E). A longitudinal nerve was visible in the dorsal side of the body wall, originating posteriorly to the nuchal organs and extending along the animal’s midline throughout the entire body length (Figure 4C,D). A pair of segmental nerves were observed to branch off from it in each segment, and in the gill segments entered the gills (Figure 4F). A thin meshwork of fibers was distributed across the entire body wall, including the dorsal side (Figure 4D).

In the region of the pygidium, 5-HT-positive fibers were found within the terminal commissure and the circumpygidial nerves (Figure 5B–E). At the site of the origin of the circumpygidial nerves, there was a remarkable branching of 5-HT-positive fibers, with fine branches entering the lobes and innervating the pygidial wall. However, no neuronal somata were identified within the pygidium itself. Additionally, within the pygidium, we found the terminal section of the intestinal plexus, from which fibers forming a network around the gut projected into the circumpygidial nerve (Figure 5C).

#### 3.1.2. FMRFamide-Positive Immunoreactivity

FMRFamide-positive cells and their processes were found in all parts of the *P. elegans* body. There were numerous neuron somata (about 50 pairs) in all areas of the brain just dorsal to the central neuropil, extending from the dorsal roots of the circumesophageal connectives up to the nuchal nerves (Figure 6A,D). We did not find any clustering of these brain neurons. They also displayed extensive processes within the neuropil and in both branches of the circumesophageal connectives (Figure 6B,C). In the area of the nuchal organs, many faintly discernible FMRFamide-positive cells were detected, mainly bipolar sensory cells. Their axons directed towards the nuchal nerves subsequently merged into the dorsal roots of the circumesophageal connectives (Figure 6E,F).

FMRFamide-positive fibers of the main palp nerve were found to arise between the ventral roots of the circumesophageal connectives and dorsal roots of the circumesophageal connectives (Figure 6A,B,I). We managed to distinguish FMRFamide-positive fibers inside an additional nerve within the palps; however, their visibility was quite feeble, and they were not consistently stained in all images. There were no FMRFamide-positive anastomoses between the main and the additional nerves. FMRFamide-positive cells were detected in the palps near the additional palp nerve but not at the base of the palp in adult animals (Figure 6I,J).

FMRFamide-positive frontal nerves of the prostomium extended anteriorly to the brain. Thin fibers from bipolar sensory cells in the upper lip region approached the frontal prostomial nerves. These cells could even display apical short cilia (Figure 6G,H and Figure 7C). Furthermore, the pharyngeal nerves approached the ventral part of the neuropil in the posterior region. Pharyngeal innervation was characterized by a high diversity of FMRFamide-positive elements. Two longitudinal pharyngeal nerves ran along the sides of the pharynx, giving rise to thinner nerves that branched towards the center, forming a network of fibers (Figure 6B and Figure 7A,B). The frontal nerves themselves bent over the upper part of the oral cavity and contributed to the pharyngeal nerves. In the anterior region, these nerves either entered the brain or formed a commissure adjacent to the brain, near the dorsal roots of the circumesophageal connectives. Additionally, FMRFamide-positive paramedian nerves in the anterior pharynx formed two commissures and two branches connecting to the dorsal roots of the circumesophageal connectives or the commissure (Figure 7B). Two pairs of ventral pharyngeal nerves arose from circumesophageal connectives closer to the subesophageal ganglion, innervated the underside of the pharynx, and had connections with the longitudinal pharyngeal nerves (Figure 7C). Notably, fine nerves originating from FMRFamide-positive sensory cells in the peristomium extended to the circumesophageal connectives (Figure 7D,E). Within the pharyngeal wall, there were thin elongated sensory cells of a triangular shape. Either individual nerve endings or long sensory processes of other cells were present in the pharynx wall (Figure 7F).

We found a pair of FMRFamide-positive unipolar cells located at the point where the ventral pharyngeal nerves emanated from the circumesophageal connectives (Figure 7C,E—yellow arrowheads). These cells possessed two processes: one extending towards the connective and the other reaching the epithelium, passing alongside several receptor cells.

The subesophageal ganglion as well as subsequent segmental ganglia contained a multitude of FMRFamide-positive neurons and exhibited well-defined commissures (Figure 8A,C–E). Segmental nerves were clearly discernible along the outgoing fibers, mainly within the nerves of the posterior segment regions. At the base of the parapodia of the first chaetiger, a pair of FMRFamide-positive unipolar neurons was identified (Figure 8B). These cells had two processes: the first one extending towards the circumesophageal connective, and the other passing posteriorly to the body wall.

We observed an FMRFamide-positive plexus within the body wall. Along the dorsal side of the body, there were longitudinal nerves, consisting of a pair of lateral nerves and an unpaired dorsal nerve (Figure 8E,G). The segmental nerves extended towards the dorsal side, forming nerve rings, where anastomoses between the nerves were distinguished. There was also one FMRFamide-positive fiber within the gill nerve (Figure 8F). Additionally, FMRFamide-positive bipolar cells, predominantly with sensory function, were present within the body wall. They could be individually dispersed throughout the body wall or aggregated in clusters, primarily localized within the parapodia and the inter-segmental regions (Figure 8G,G’).

The midgut contained a plexus of multidirectional fibers and individual cell somata (Figure 7G,H). We identified two types of cells: bipolar neurons in the plexus, which extended their processes over long distances, and triangular sensory cells in the intestinal epithelium. The hindgut hosted a notably abundant population of FMRFamide-positive cells, forming a basket-like structure (Figure 8H,I). Many unipolar cells were concentrated around the anus. In the pygidium itself, each lobe contained a single sensory bipolar cell, with its processes extending towards the circumpygidial ring. FMRFamide-positive fibers within the terminal commissure were scarce.

### 3.2. Regeneration of the Nervous System

#### 3.2.1. 5-HT-Positive Nervous System Regeneration

Immediately following amputation, some regenerates had an everted gut, resulting in a mildly “divergent” appearance of the ventral nerve cord (Figure 9A and Figure 10A). However, at the amputation site, we only found mechanical tissue damage, contraction of the wound tissue, and the associated deformations of the nervous system. At 4 hpa the pattern was almost the same in posterior and anterior regenerates (Figure 10B). At 12 h after amputation, the wound began to tighten. The terminal ends of the ventral nerve cord looked frayed, which was evident in both acetylated α-tubulin-positive and 5-HT-positive fibers (Figure 9B and Figure 10C). Frayed ends of the ventral nerve cord became more pronounced at 24 h after amputation (Figure 9C and Figure 10D). There was some tendency towards an increased number of varicosities in the terminal fibers of the ventral nerve cord at the amputation site in the posterior regenerate (Figure 10D). Notably, single 5-HT-positive fibers in the ventral nerve cord began to grow into adjacent tissues in the anterior regenerate starting from 24 hpa (Figure 9C).

Evident indications of fiber growth emerged at two days after amputation both in the posterior and the anterior regenerates (Figure 9D and Figure 10E). These were single fibers extending from damaged connectives of the ventral nerve cord into the regenerating blastema. By the third day of regeneration, a considerable number of 5-HT-positive elements were observed in the anterior regenerate, including elements of the ventral nerve cord, the brain, and the peripheral nervous system (peripheral nerves of the body wall) (Figure 9E). At the posterior regeneration site, 5-HT-positive elements of the ventral nerve cord grew into the newly formed segments. By this point, the pygidium was already developed in the posterior regenerate, yet we did not observe the growth of 5-HT-positive fibers there (Figure 10F). On the fourth day of regeneration, there was an increase in the number of 5-HT-positive elements in the brain (Figure 9F). 5-HT-positive fibers became evident in the palp buds. Several newly formed segments (2–3 segments) were formed in the posterior regeneration site, and the pygidium had all the characteristic structures found in an intact posterior end. On the ventral side, we found newly formed nerves of pygidial lobes (Figure 10G). On the dorsal side, 5-HT-positive fibers of peripheral nerves of the body wall in the posterior regenerate were visible (Figure 10H).

On the 5th day of regeneration, we observed no significant changes in either the anterior or the posterior regenerates (Figure 9G). However, on the 6th day of regeneration, the dorsal nerve began to grow from the intact segment in the anterior regenerate (Figure 9H). It should be noted that the dorsal nerve was labeled with acetylated α-tubulin antibody all along the anterior regenerate; however, 5-HT-positive fibers were found only in its proximal part. On the 7th day of regeneration, 5-HT-positive cell somata within the brain and segmental ganglia became discernible in the anterior regenerate, and peripheral nervous system details, such as segmental nerves, appeared throughout the entire regenerate (Figure 9I,J). Notably, the somata were also clearly visible in the posterior regenerate but were smaller than in the intact part (Figure 10I). By the 9th day of regeneration, the structure of the 5-HT-positive nervous system in both the anterior (Figure 9K,L) and the posterior regenerates closely resembled that of intact worms. The regenerates differed from the intact parts in slightly smaller sizes.

#### 3.2.2. FMRFamide-Positive Nervous System Regeneration

Immediately after amputation, only changes associated with the surgical procedure were observed (see above) (Figure 11A and Figure 12A). Starting from 4 h post-amputation, the connectives of the ventral nerve cord appeared frayed, and an increase in the FMRFamide concentration at the amputation site was observed (Figure 11B). All these events became more pronounced at 12 h post-amputation in both the anterior (Figure 11C) and the posterior (Figure 12B,C) regenerates. At 24 h after amputation, single FMRFamide-positive fibers from the ventral nerve cord initiated prominent growth into the anterior regenerate (Figure 11D,D’) and a moderate growth into the posterior regenerate (Figure 12D).

By the 2nd day post-amputation, the growth of FMRFamide-positive fibers in the anterior and the posterior regenerates was clearly visible in all examined specimens (Figure 11E and Figure 12E). There was also a network of fibers along the gut in the posterior regenerate. On the 3rd day after amputation, a significant increase in the number of FMRFamide-positive elements was observed in the growing head (Figure 11F) compared to the 2 dpa. These FMRFamide-positive fibers were present in the ventral nerve cord, the brain primordium, and peripheral gut nerves. On the 4th day of regeneration, no significant changes occurred in the anterior regenerate (Figure 11G). On days 3–4 after amputation, the connectives of the ventral nerve cord and the meshwork around the gut progressively extended along the growing tail (Figure 12F,G). However, the number of fibers within the growing ventral nerve cord remained relatively small compared to the intact region. On the 5th day of regeneration, we observed a slight increase in the number of nerve elements within both the anterior and the posterior regenerates (Figure 11H). Then, on the 6th day of regeneration, the initial FMRFamide-positive cell somata appeared in the growing head, and fibers began to grow into the developing palps (Figure 11I).

By the 7th day of regeneration, in the growing head, FMRFamide-positive elements were identified across all major brain regions (Figure 13A). Their presence was accompanied by a significant increase in the number of somata in the brain (Figure 13B). Additionally, the pharyngeal nerves became visible in the newly formed pharynx, and the innervation of the entire intestine was distinctly discernible (Figure 13A,D,E). However, in the segmental ganglia, only the connectives of the ventral nerve cord were present, while neuronal somata were lacking (Figure 13C). In the posterior regenerate on the 7th day post-amputation, FMRFamide-positive fibers were prominently visible in the ventral nerve cord, although cell somata had not yet emerged (Figure 12I). The intestinal innervation was developed almost like that of an intact worm (Figure 12J). Along the dorsal side, the dorsal longitudinal nerve and segmental nerves could be observed throughout the entire length of the growing tail, and cell somata in the parapodia were also apparent (Figure 12H,J). By the 9th day of regeneration, the neuronal somata in the brain and the segmental ganglia were developed in the anterior regenerate (Figure 13F–H). The peripheral innervation of the dorsal side of the regenerate closely resembled that of intact worms. Additionally, cells in the palps became visible (Figure 10G,G’). In the posterior regenerate, most of the structures were already present, as in intact segments. However, the somata within the segmental ganglia and the commissures remained somewhat indistinct at this stage (Figure 12K).

## 4. Discussion

### 4.1. The General Structure of the Nervous System

Our data on the 5-HT- and FMRFamide-positive nerve elements contribute to the knowledge of the structure and regeneration of the *P. elegans* nervous system. Previously, we studied the nervous system of adult and regenerating worms using histochemical detection of catecholamine (CA)-containing cells [10,45] and immunohistochemical labeling of histamine (HA) and gamma aminobutyric acid (GABA) [43]. Here, we compare all our previous data on *P. elegans* with the information available on the structure and regeneration of the nervous system in other annelids.

#### 4.1.1. Central Nervous System (CNS)

We provided a comprehensive description of the distribution of 5-HT- and FMRFamide-positive elements in the body of *P. elegans*. In broad terms, the overall structure of the nervous system corresponds to that in other annelids [19,46,47]. The general brain organization visualized with antibodies conforms to the descriptions of Orrhage for other spionids [46,48]. The ventral nerve cord (VNC) exhibits a distinct segmental organization of ganglia, with a repeating pattern of neuron distribution, commissures, and segmental nerves. This is consistent with the previous studies. Regarding the quantification of the identified neuronal somata, FMRFamide is notably more abundant than 5-HT. This discrepancy may be associated with the fact that the antibodies can detect various FMRFamide-like peptides, rather than a single specific one [23]. 5-HT-positive cell somata were predominantly located in the CNS. Some somata were also present in the stomatogastric system, and a few were observed within the palps (see below). The processes of 5-HT-positive cells were found throughout the entire body (Figure 14A–E). This pattern of distribution of 5-HT-positive immunoreactivity is typical for annelids [19,46,49]. FMRFamide-positive neuronal bodies were found not only in the CNS but also in the peripheral nervous system (PNS), indicating a broad range of functions of this neuropeptide (Figure 15A–E).

#### 4.1.2. The Palps and Nuchal Organs

The palps play a significant role as sensory organs and are involved in feeding behavior. Palp nerves consist of afferent sensory and efferent motor fibers [50]. The innervation pattern in the palps recorded in our study is generally consistent with earlier descriptions [46,47]. Bringing together our previous data on various neurotransmitter distributions, we can reconstruct a complex and diverse innervation in the palps [43,45]. All studied neuromediators were found in the main palp nerve. CA-, 5-HT-, and FMRFamide-positive elements were found in several additional nerves. It should be noted that the additional nerves of the palps anastomosed and passed into each other. In receptor cells CA-, HA-, GABA-, and 5-HT-positive immunoreactivity was found. GABA- and FMRFamide-positive immunoreactivity was detected in additional non-receptor cell types. Round cells, without processes, near the main nerve showed GABA-positive immunoreactivity, while those near additional nerves showed FMRFamide-positive immunoreactivity. A pair of GABA-positive cells were also found at the base of the palps. Thus, our data indicate a complex and neurodiverse structure of the palp as a sensory organ. The cells described in our study have various of shapes and positions and evidently perform different sensory and secretory functions.

The ciliated structures on the spionid palp can be classified based on their positions as frontal, latero-frontal, lateral, and abfrontal [50,51]. The putative functions of these cilia types are as follows: lateral and abfrontal cilia are presumed to be chemoreceptors, while the latero-frontal cilia are considered as strong candidates for the role of mechanoreceptors. 5-HT immunoreactivity in the palps has been observed in several palp nerves and cells underlying the food groove of *P. elegans* and *Dipolydora quadrilobata* [50]. However, based solely on confocal images, it is difficult to determine the spatial positioning of cells in the palps. In our study, we did not find any 5-HT-positive cells underlying the food groove. According to our data, the location of 5-HT-positive receptor cells of the palps is more likely to be of the latero-frontal type. If so, these cells can perform a mechanoreceptor function.

According to Forest and Lindsay [50], some somata of abfrontal palp receptor cells and palp nerves underlying the food groove have FMRFamide-positive immunoreactivity. The FMRFamide-positive cells that we found in the palps are similar to these abfrontal cells; however, we were unable to detect their processes. In the course of head regeneration, FMRFamide-positive somata in the palps were not detected until the 9th day of regeneration. Ciliated cells of the lateral and abfrontal surfaces of the palp have always been considered chemosensory, according to their morphology and cell labeling results [51,52]. Thus, FMRFamide-positive immunoreactive cells apparently have a chemosensory nature (primary olfactory neurons).

Ultrastructural data have revealed that the nuchal organs of *P. elegans* are represented by ciliated cells and bipolar primary sensory cells [53]. The afferent innervation of these sensory cells is associated with the nuchal nerves, which extend to the dorsal roots of the circumesophageal connectives. Among all the neurotransmitters examined, only FMRFamide-positive receptor cells were found in the nuchal organs. Similar cells displaying FMRFamide-positive immunoreactivity were also identified in the upper lip of the worm. These findings are consistent with those reported for *D. quadrilobata*, whose sensory cells in the sensory organs also exhibit FMRFamide-positive immunoreactivity [50]. Therefore, we can assume the involvement of FMRFamide in chemoreception in spionids.

#### 4.1.3. Innervation of the Pharynx

In the pharynx, both 5-HT and FMRFamide were detected in paired longitudinal pharyngeal nerves. Innervation of the pharyngeal region is illustrated in Figure 16. These nerves run along the lateral sides of the pharynx and connected with each other forming a loop at its ventral border with the midgut. Additionally, thin nerves were found to arise from the pharyngeal nerves that entwined the pharynx above and below. At the anterior part of the pharynx, a relatively thick commissure was observed between these nerves. However, we did not find any 5-HT- or FMRFamide-positive neurons in this area. Nevertheless, in the posterior pharynx, near the pharyngeal loop, we did find individual paired FMRFamide-positive neurons. This finding raises the question about the existence of separate esophageal ganglia in *P. elegans*. Ganglia of this kind were found in other annelids, but in species with a more complex pharyngeal organization [45,54,55,56]. Further studies of pharyngeal innervation are needed to resolve this problem.

The pharyngeal nerves are a direct extension of the frontal nerves, which are the stomatogastric nerves originating from the anterior part of the brain in Orrhage’s terminology [46]. In this context, we deviate from the Orrhage terminology and refer to the anterior stomatogastric nerves as frontal nerves. This nomenclature shift is justified by the fact that they innervate not only the pharynx and the upper part of the oral cavity but also the entire anterior part of the prostomium. This fact has been well substantiated for catecholamines [45]. The innervation of the pharynx in *P. elegans* also involves two pairs of nerves that branch from the circumesophageal connectives at the ventral side of the peristomium and run through the lower lip to the pharynx. A somewhat comparable, although more complex, innervation pattern has been observed in representatives of the Eunicida [37,56,57] supporting the idea that the ventral pharynx and dorsolateral ciliary folds may be the plesiomorphic condition for Annelida [58,59]. However, to provide a comprehensive analysis, more detailed studies of the innervation of the pharyngeal region in different annelid groups are required.

### 4.2. Regeneration

In our previous studies, we have described the morphological stages of *P. elegans* regeneration [10], which can be summarized into the following main stages: wound healing, blastema, the primordium of the head and pygidium, segmentation, differentiation, and growth. These stages correspond to those in the regeneration of other annelid species [35]. Here, we compare data on the regeneration of 5-HT-positive and FMRFamide-positive elements with previous results on CA-, GABA-, and HA-positive elements [10,43]. The results are summarized in Table 1 in accordance with the morphological stages of *P. elegans* regeneration.

#### 4.2.1. Anterior Regeneration and the Central Nervous System (CNS)

Early stages of the nervous system regeneration in annelids of different families have many similarities [20,28,30,38]. At the wound healing stage and later in the blastema in *P. elegans*, the first 5-HT-positive and FMRFamide-positive thin fibers began to grow from the intact part of the VNC (Figure 17A and Figure 18A). In *Timarete* cf. *punctata* and *Typosyllis antoni*, a plexus was found in the blastema at this stage. This plexus later transformed into the neuropil of the cerebral brain and VNC [20,28]. This differs from the other studied species including *P. elegans*. Two explanations of this phenomenon may be suggested. The first is that the formation of plexus in the blastema may be related to species- or taxon-specific characteristics in the morphogenesis of regeneration. An alternative explanation is associated with variation in the degree of centralization of the nervous system in annelids. In *P. elegans*, at early regeneration stages, we observed that the damaged ends of VNC had a frayed look. Perhaps in *T.* cf. *punctata* and *T. antoni* the disintegration of the nerve cord is more pronounced because of a higher centralization of the VNC and a greater number of fibers in it, which results in the need of a deeper rearrangement during regeneration.

In *P. elegans*, the scaffold of acetylated α-tubulin-, 5-HT-, and FMRFamide-positive processes began to develop in the patterning blastema from the 3rd day after ablation (Figure 17C and Figure 18C). These fibers formed two bundles that extended through the regenerate and connected in the head primordium, to form a loop-like structure. This loop is represented by a pair of circumesophageal connectives, passing into cerebral commissures. Ozpolat and Bely [35] paid special attention to this stage of the brain primordium. The presence of a loop has been observed in different groups of annelids. Some species even display even multiple loops, with the central loop corresponding to the cerebral commissure, and lateral ones to the dorsal roots of the circumesophageal connectives [28,60]. Previous studies have also identified the presence of dorsal roots of the circumesophageal connectives at the single loop stage in the blastema of *D. quadrilobata* and *P. elegans* [38]. However, our results suggest that the neurites cannot be clearly assigned to the dorsal or ventral roots of the circumesophageal connectives at this stage of regeneration.

The first 5-HT- and FMRFamide-positive nerve fibers appeared in an early blastema of *P. elegans*. These fibers originated from the intact VNC and scaffolded the new VNC and brain primordium at the blastema patterning stage. By this time, it was still impossible to distinguish the cell somata in the brain primordium and the segmental ganglia. The cell somata of FMRFamide- and 5-HT-positive immunoreactivity began to emerge by the 6–7th days of regeneration. However, other neuroactive substances may show different dynamics. For example, the first GABA-positive neurons appeared in the brain primordium by the 4th day of regeneration, despite the fact that GABA-positive fibers from the intact VNC have not yet reached the head primordium [43]. In brain extirpation experiments on *Lumbricus terrestris* and *Eisenia fetida*, early appearance of fibers and cell somata has also been observed in the developing cerebral ganglion [39,42,61]. By the 3rd day of regeneration, 13 FMRFamide-positive neurons and some GABA-positive neurons were found in the blastema in *E. fetida.* In *L. terrestris*, some 5-HT-positive neuronal somata were identified at this stage. The authors emphasize a significant regulatory role of 5-HT, FMRFamide, and GABA in cell migration and proliferation in neurogenesis. This suggestion is indirectly confirmed by the earlier appearance of these neurotransmitters during both regeneration and in embryonic development of annelids and mollusks [62,63,64,65,66,67]. In the course of *P. elegans* regeneration, we first observed only the processes, while neuronal somata became visible later. It is plausible that the differentiation of neurons occurs before their actual detection since the antibody labeling reveals specific substances that may appear in the cell only at the later stages of its differentiation. Further comparative studies on the early stages of neuronal differentiation during the regeneration process will shed light on this problem.

Regeneration shares certain similarities with embryogenesis; hence, it is frequently considered a common model for developmental processes. Typically, the parameters used for comparison involve stages of developmental programs and patterns. For instance, Weidhase [28] emphasizes similarities in the simultaneous formation of three segments in the anterior regenerate of *Timarete* cf. *punctata* and its larval development. However, pelagic annelid larvae are usually active self-sufficient organisms, sometimes with larva-specific structures. The larva undergoes a gradual increase in complexity during development, and the formation of the nervous system may be prolonged up to the juvenile stage. In contrast, during regeneration, it is critically important that all adult body structures are restored rapidly and completely. Perhaps regeneration may have even more extensive similarities wint direct development or fission (as in some oligochaetes) [30,66].

During the regeneration of the nervous system in *P. elegans*, neuronal elements with different chemical specificity appear at different times (Table 1). Thus, the following question arises: How similar are the processes of development and regeneration in this respect? In annelids and mollusks, 5-HT-, FMRFamide-, GABA-, and dopamine-positive cells were identified as pioneer elements in neurogenesis that form a scaffold for the further formation of the nervous system. Unfortunately, there are no data available on the development of the nervous system in *P. elegans* during embryogenesis. Poecilogony and development in tubes [8,11], make it difficult to obtain larval material for the analysis of developmental processes. However, the early development of another spionid species, *Malacoceros fuliginosus* has been recently studied [21]. In this species, the first pioneer posterior and anterior neurons have been identified at 9 hpf and 16 hpf. Nonetheless, these neurons are associated with an unknown neurotransmitter. Peripheral sensory cells labeled with FMRFamide and 5-HT appear in the larva of *M. fuliginosus* at 14 hpf and 21 hpf, respectively. Thus, there seems to be considerable variation in the chemical specificity of the first neural elements involved in regeneration and development among different animals. The timing of pioneer nervous element appearance in various organisms may be influenced by external factors (food availability, breeding season, external stress, abiotic environmental conditions, etc.) and internal ones (type and possibilities of regeneration, regulatory elements, etc.) [42,61,66,68,69]. It is likely that patterning mechanisms and the expression of the same genes have some similarities during regeneration and during development of organisms [35].

#### 4.2.2. Posterior Regeneration

In the posterior regenerate of *P. elegans*, the first nerve elements became visible only by the 2nd day of regeneration (Figure 17B and Figure 18B). However, the early stages of neuroregeneration were similar in both the anterior and the posterior end. The differences in regeneration between the anterior and the posterior regenerate became more pronounced at later stages [35]. Moreover, it has been demonstrated in the Syllidae that in terms of regulatory mechanisms and gene expression, posterior regeneration is comparable to normal growth [29]. In *P. elegans*, there are also differences in the growth patterns of the anterior and the posterior regenerates; however, the total time of regeneration is comparable (just over a week). At the posterior end, after the pygidium formation, the growth zone was activated by 3–4 days of regeneration, leading to the sequential formation of new segments (Figure 17D,F and Figure 18D,F). 5-HT-positive nerves of pygidial lobes and peripheral nerves in the body wall were discovered by the 4th day of regeneration. The results of our previous studies indicate that CA-positive nerve elements of the pygidium are also visible by the 4th day of regeneration [10]. By this time, the pygidium is morphologically well-developed. FMRFamide-positive elements in the pygidium have not yet appeared even by the 9th day of regeneration. Thus, despite the early initiation of FMRFamide fiber growth, it requires a long time for all FMRFamide elements to develop during neuroregeneration.

#### 4.2.3. Regeneration of the Peripheral Nervous System (PNS)

The majority of research on annelid nervous system regeneration centers on CNS redevelopment, while regeneration of the PNS is rarely the primary focus [20,28,30,36,37,38,70]. PNS components, such as palp nerves or segmental nerves are usually just mentioned in passing [20,28]. However, there are examples of strikingly unusual regeneration of the PNS. In the oligochaetes *Pristina leidi*, *Enchytraeus fragmentosus*, and *Stylaria lacustris*, nerve fibers branch off from intact segmental nerves and innervate the blastema [30,36]. In *P. elegans*, we identified 5-HT- and FMRFamide-positive fibers and receptor cells of PNS at different stages of regeneration. The innervation of the intestine by FMRFamide-positive elements started from the intact intestine and extended to the patterning blastema (Figure 18A–H). The 5-HT-positive peripheral nerves in the body wall and nerves of the pygidial lobes grew from the newly formed VNC by the 4th day of regeneration (Figure 17G,H). The 5-HT-positive fibers of the dorsal nerve began to grow into a regenerate from the intact part of the nerve only by the 6th day of regeneration (Figure 17E,G). In contrast, by the 3rd day of regeneration, the CA-positive dorsal nerve grew from the intact part and branched in the anterior regenerate [10]. CA-positive receptor cells could be distinguished at the distal end of the blastema. They increased in number during regeneration. CA-positive nerve elements in the pygidium and segmental nerves of the posterior regenerate were detected at 4 dpa and 7 dpa, respectively. We suggest that the PNS emerges at the early stages of regeneration almost simultaneously with the CNS. As shown previously, the stages of CNS regeneration are similar even within such a large and diverse group as Annelida [35]. The distribution and regeneration patterns of the PNS may depend more on the individual features of the animal related to musculature or sensory organ development.

#### 4.2.4. Palp and Food Groove Regeneration

Studies of regeneration usually focus on the morphological aspects [2,28,30,36], while the functional aspects often remain unexplored. Therefore, the most accessible approach to estimate functionality is to examine the ability of an animal to feed. *P. elegans* collects food particles from the sediment surface with the help of the palps [32,38,71]. When do regenerating palps become functional for feeding? The functionality of the palps depends on the development of the food groove. During regeneration the food groove grows gradualy from the distal end of the palp to its base. The palps are likely to become functional when the nervous system components within them are reconstituted.

According to our data, the first 5-HT-positive nerve fibers began to grow into the palp buds by the 4th day of regeneration (Figure 17E). At this stage, the palps were short and lacked cilia. By the 6–7th day of regeneration, the main and additional palp nerves became distinguishable as well as some of the receptor cells, primarily at the distal end. The ciliary groove was developed in 1/3 of the total length of the palp on its distal end. The development and growth direction of the food groove corresponded to the appearance of the nerve cells in the palps. By the 7th day of regeneration, the innervation of the palp was quite diverse, with all the nerves and some of the cells having been formed (Figure 17G and Figure 18G). However, it has been demonstrated in experiments that due to the limited development of the food groove (30%) by this point, feeding remains impossible [38]. Furthermore, our histological data also indicated that feeding was still impossible because the pharynx and the midgut only fused into a single system by 7 dpa (our unpublished data). The FMRFamide-positive cells appeared in the palps by the 9th day of regeneration. At this stage, the palps possessed a well-developed nervous system and were probably fully functional.

## 5. Conclusions

We showed that 5-HT- and FMRFamide-positive elements in the *P. elegans* nervous system were represented by diverse structures, whoseregeneration followed a certain pattern. We identified the following stages of CNS neurogenesis during regeneration (1) single nerve fibers extending from the intact VNC into the wound tissue and the blastema; (2) formation of a loop as the primordium of the cerebral ganglion; and (3) the appearance of neuron somata in the brain and the segmental ganglion of the differentiating regenerate. Elements of the PNS could be traced at different stages of regeneration starting from the blastema. Newly formed PNS fibers originated from the intact part of the PNS or emerged from the newly formed sections of the CNS.

The following conclusions were drawn after a comparison with previous results (Table 1):

(1) 5-HT and FMRFamide are the first neurotransmitters in the regeneration of the CNS; CAs are the first neurotransmitters in the regeneration of the PNS;

(2) The regeneration of nerve elements with different chemical specificities proceeds at different speeds and has a different duration: 5-HT and CA being early and fast; FMRFamide, early and slow; GABA, late and fast; and HA, late and slow.

## Figures and Tables

**Figure 1 biology-12-01412-f001:**
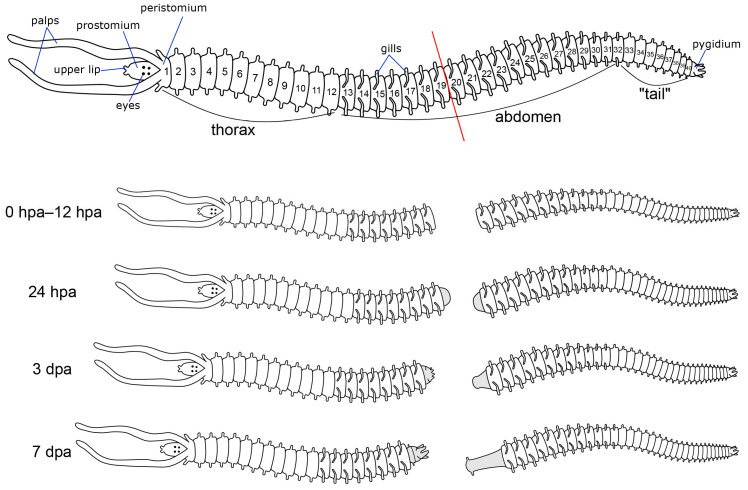
Schematic diagram of the experimental procedure. The red line indicates the cutting plane.

**Figure 2 biology-12-01412-f002:**
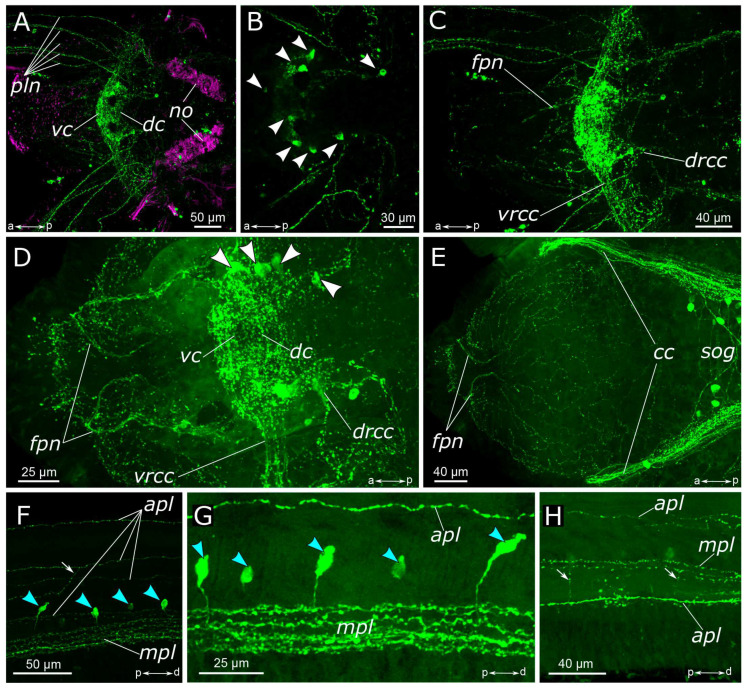
5-HT-positive elements in the prostomium and palps of *Pygospio elegans*. (**A**–**D**) Partial z-projections through the cerebral ganglion in the dorsoventral direction. (**E**) Innervation of the ventral side of the prostomium and peristomium. (**F**–**H**) 5-HT-positive elements of the palps. Abbreviations: white arrowheads indicate neurons; blue arrowheads indicate immunopositive cells in the palps; white arrows indicate anastomoses of 5-HT-positive nerves; *apl*—additional palp nerve; *cc*—circumesophageal connectives; *dc*—dorsal cerebral commissure; *drcc*—dorsal roots of the circumesophageal connectives; *fpn*—frontal prostomial nerve; *mpl*—main palp nerve; *no*—nuchal organ; *pln*—palp nerve; *sog*—subesophageal ganglion; *vc*—ventral cerebral commissure; *vrcc*—ventral roots of the circumesophageal connectives; 5-HT-positive elements—green; and acetylated α-tubulin-like elements–magenta. Double-sided a-p arrows in (**A**–**E**) indicate the direction of the anteroposterior axis. Double-sided p-d arrows in (**F**–**H**) indicate the direction of the proximodistal axis.

**Figure 3 biology-12-01412-f003:**
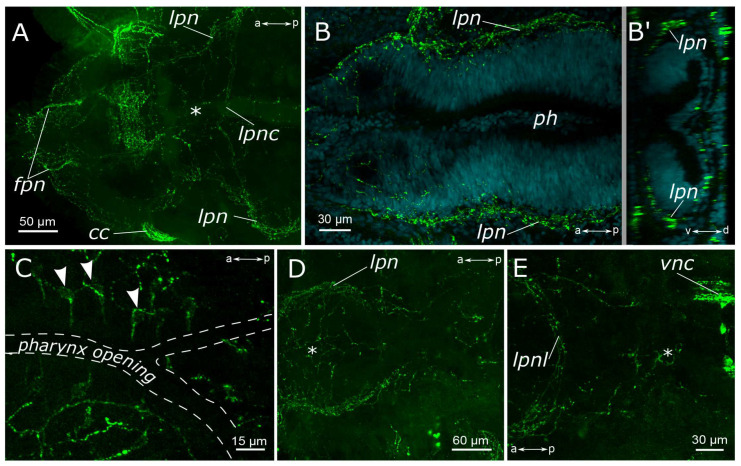
5-HT-positive elements of the foregut of *Pygospio elegans*. (**A**,**B**) Innervation of the pharynx. (**B’**) Virtual cross-section of the pharynx. (**C**) Sensory cells of the pharyngeal epithelium. (**D**,**E**) Innervation of the proximal part of the gut and pharynx. Abbreviations: asterisk indicates nerve plexus; white arrowheads indicate sensory cells; *cc*—circumesophageal connectives; *fpn*—frontal prostomial nerve; *ip*—intestinal plexus; *lpn*—lateral pharyngeal nerves; *lpnc*—lateral pharyngeal nerve commissure; *lpnl*—lateral pharynx nerve loop; *ph*—pharynx; and *vnc*—ventral nerve cord. 5-HT-positive elements—green; DAPI–cyan. Double-sided a-p arrows indicate the direction of the anteroposterior axis. Double-sided d-v arrow in (**B’**) indicates the direction of the dorsoventral axis.

**Figure 4 biology-12-01412-f004:**
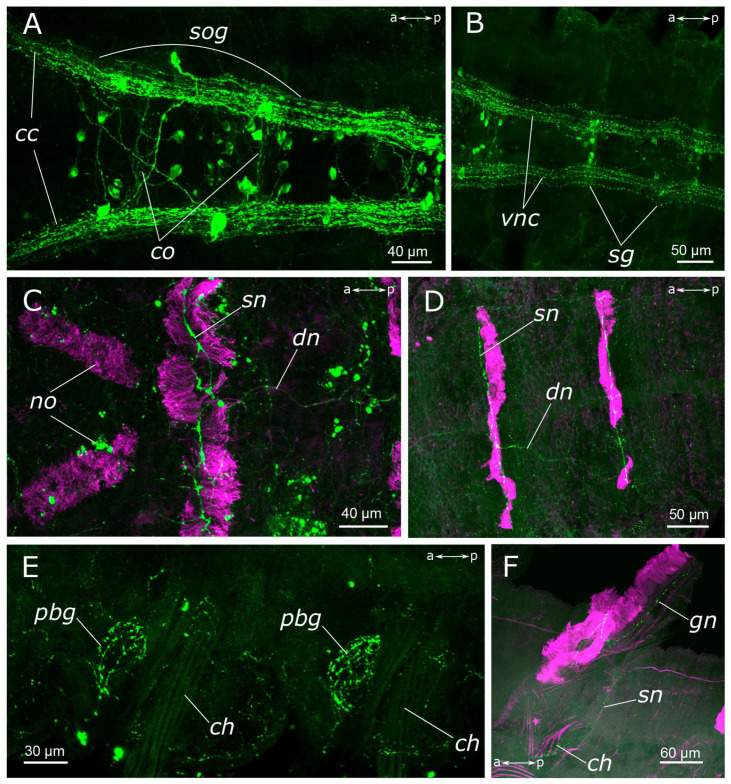
5-HT-positive elements in body segments of *Pygospio elegans.* (**A**–**D**) Innervation of thoracic segments (**A**,**B**) subesophageal and first segmental ganglia, ventral view, (**C**,**D**) first thoracic segments, dorsal view. (**E**) Innervation of glands at the parapodial bases. (**F**) Gill innervation. Abbreviations: *cc*—circumesophageal connectives; *ch*—chaetae; *co*—commissure of the segmental ganglion; *dn*—dorsal longitudinal nerve; *gn*—gill nerve; *no*—nuchal organs; *pbg*—parapodial base glands; *sg*—segmental ganglion; *sn*—segmental nerve; *sog*—subesophageal ganglion; *vnc*—ventral nerve cord; 5-HT-positive elements—green; and acetylated α-tubulin-like elements—magenta. Double-sided a-p arrows in (**A**–**E**) indicate the direction of the anteroposterior axis.

**Figure 5 biology-12-01412-f005:**
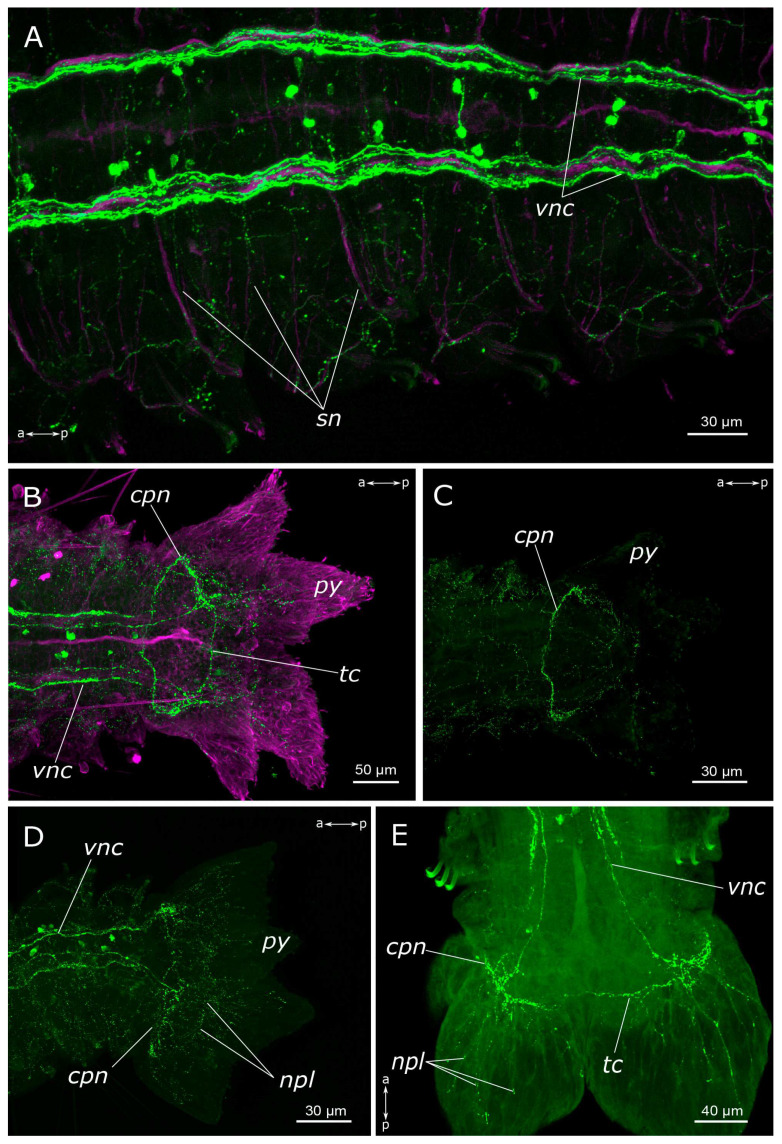
5-HT-positive elements of abdominal segments and the pygidium of *Pygospio elegans*. (**A**) Innervation of abdominal segments. (**B**–**E**) Pygidium innervation (**B**,**D**,**E**) ventral side, (**C**) dorsal side. Abbreviations: *cpn*—circumpygidial nerve; *npl*—nerves of pygidial lobes; *py*—pygidium; *sn*—segmental nerves; *tc*—terminal commissure; *vnc*—ventral nerve cord; 5-HT-positive elements—green; and acetylated α-tubulin-like elements—magenta. Double-sided a-p arrows indicate the direction of the anteroposterior axis.

**Figure 6 biology-12-01412-f006:**
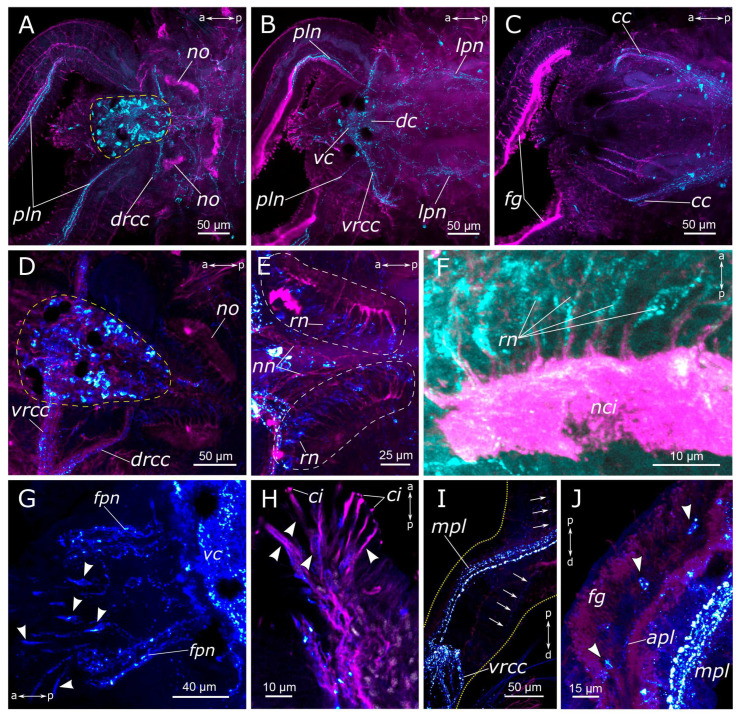
FMRFamide-positive elements in the prostomium and palps of *Pygospio elegans*. (**A**–**C**) Partial z-projections through the cerebral ganglion in the dorsoventral direction. (**D**) Cerebral ganglion and its projections. (**E**) Innervation of the nuchal organs. (**F**) Receptor cells of the nuchal organ. (**G**) Innervation of the frontal part of the prostomium. (**H**) Prostomium receptor cells. (**I**,**J**) Palp innervation. Abbreviations: white arrowheads indicate immunopositive cells; white arrows indicate an additional palp nerve; white dashed lines indicate the nuchal organ; the cerebral ganglion is outlined by a yellow dashed line; yellow dotted lines outline the palp; *apl*—additional palp nerve; *cc*—circumesophageal connectives; *ci*—cilia; *dc*—dorsal cerebral commissure; *drcc*—dorsal roots of the circumesophageal connectives; *fg*—food groove; *fpn*—frontal prostomial nerve; *lpn*—lateral pharyngeal nerves; *mpl*—main palp nerve; *nci—*nuchal cilia; *nn*—nuchal nerve; *no*—nuchal organ; *pln*—palp nerve; *rn*—receptor cell of the nuchal organ; *vc*—ventral cerebral commissure; and *vrcc*—ventral roots of the circumesophageal connectives. FRMFamide-positive elements—cyan and turquoise; acetylated α-tubulin-like elements—magenta. Double-sided a-p arrows in (**A**–**H**) indicate the direction of the anteroposterior axis. Double-sided p-d arrows in (**I**,**J**) indicate the direction of the proximodistal axis.

**Figure 7 biology-12-01412-f007:**
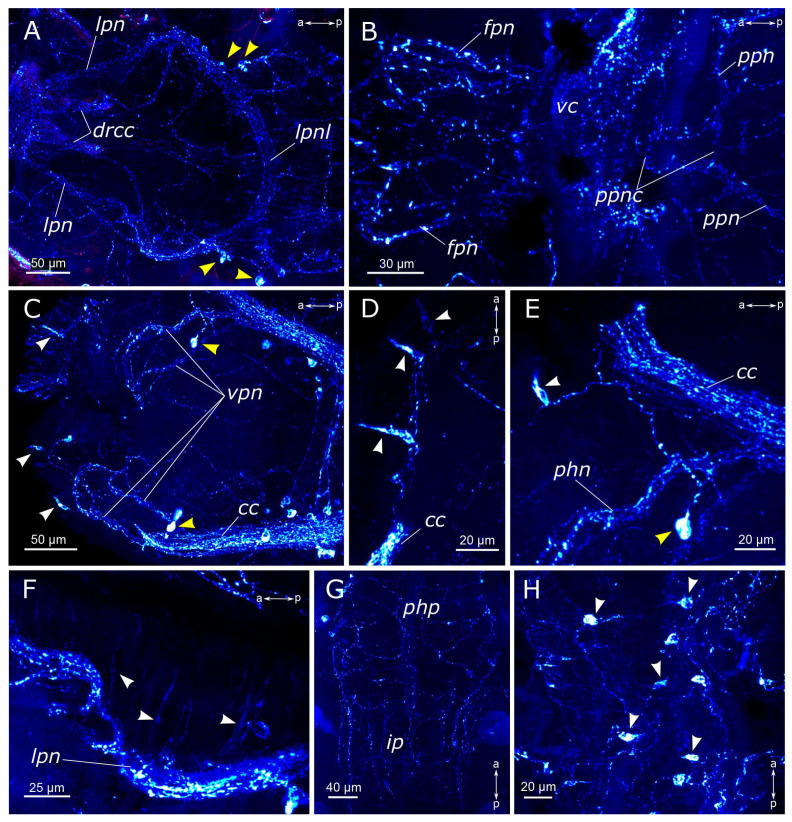
FMRFamide-positive elements of the foregut of *Pygospio elegans*. (**A**–**C**) Foregut innervation. (**D**) Receptor cells of the mouth region. (**E**) Neuron of circumesophageal connective. (**F**) Sensory cells of the pharynx wall. (**G**) Innervation of the pharynx and intestinal region. (**H**) Receptor cells of the gut wall. Abbreviations: white arrowheads indicate receptor cells; yellow arrowheads indicate neurons innervating the pharynx (and gut); *cc*—circumesophageal connectives; *drcc*—dorsal roots of the circumesophageal connectives; *fpn*—frontal prostomial nerve; *ip*—intestinal plexus; *lpn*—lateral pharyngeal nerves; *lpnl—*lateral pharynx nerve loop; *php*—pharyngeal plexus; *ppn*—paramedian pharyngeal nerves; *ppnc*—paramedian pharyngeal nerve commissure; *vc*—ventral cerebral commissure; *vpn*—ventral pharyngeal nerve; and FMRFamide-positive elements—cyan. Double-sided a-p arrows indicate the direction of the anteroposterior axis.

**Figure 8 biology-12-01412-f008:**
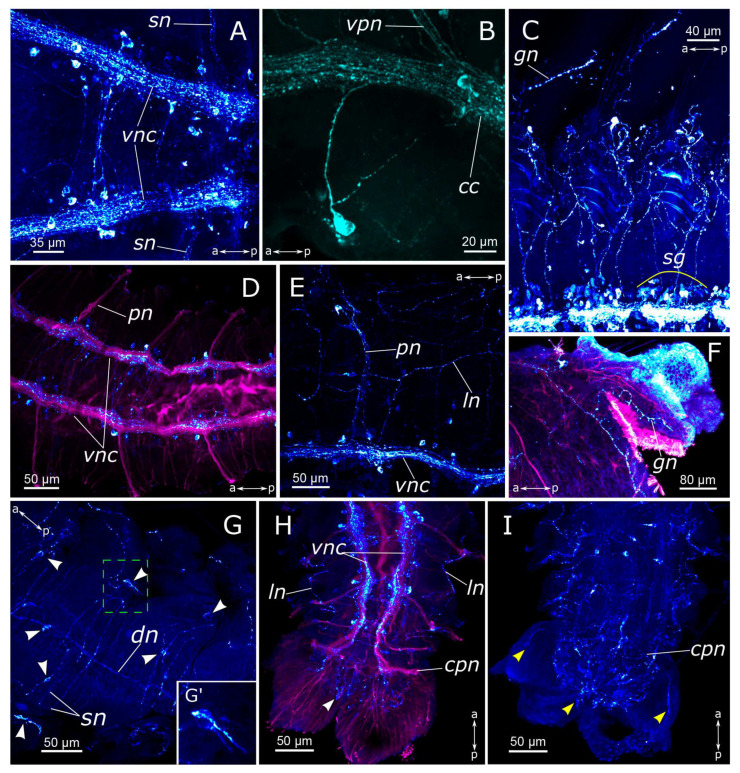
FMRFamide-positive elements in body segments and the pygidium of *Pygospio elegans*. (**A**) Subesophageal ganglion. (**B**) A neuron at the base of the first chaetiger’s parapodia. (**C**–**E**) Ganglia of abdominal segments and innervation of the body wall. (**F**) Gill innervation. (**G**) Body wall innervation (dorsal view). (**G’**) Receptor cell of the body wall. (**H**,**I**) Pygidium innervation. Abbreviations: white arrowheads indicate receptor cells of the body wall; yellow arrowheads—pygidial receptor cells; *cc*—circumesophageal connectives; *cpn*—circumpygidial nerve; *dn*—dorsal longitudinal nerve; *gn*—gill nerve; *ln*—lateral longitudinal nerve; *pn*—parapodial nerve; *sg*—segmental ganglion; *sn*—segmental nerve; *vnc*—ventral nerve cord; *vpn*—ventral pharyngeal nerve; FMRFamide-positive elements—cyan; acetylated α-tubulin-like elements—magenta. Double-sided a-p arrows indicate the direction of the anteroposterior axis.

**Figure 9 biology-12-01412-f009:**
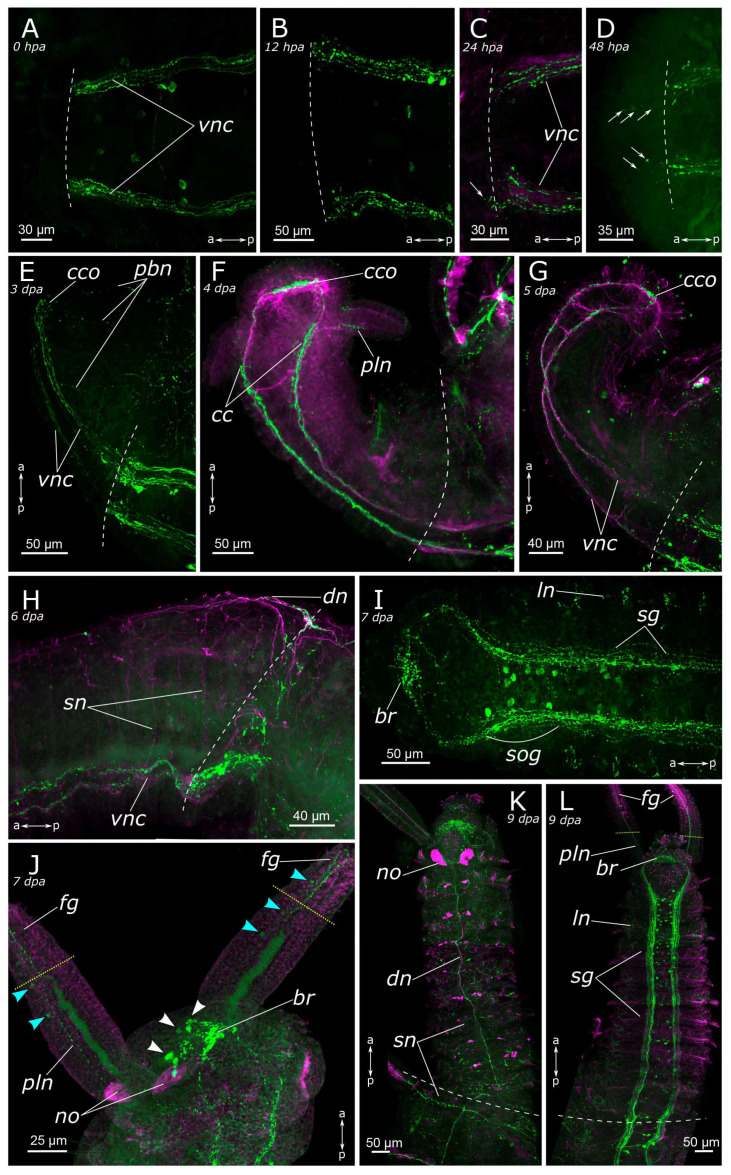
Anterior regeneration of the 5-HT-positive elements of the *Pygospio elegans* nervous system. (**A**) 0 hpa, ventral view. (**B**) 12 hpa, ventral view. (**C**) 24 hpa, ventral view. (**D**) 48 hpa, ventral view. (**E**) 3 dpa, ventro-lateral view. (**F**) 4 dpa, ventro-lateral view. (**G**) 5 dpa, ventro-lateral view. (**H**) 6 dpa, lateral view. (**I**) 7 dpa, ventral view. (**J**) 7 dpa, ventro-lateral view. (**K**) 9 dpa, dorsal view. (**L**) 9 dpa, ventral view. Abbreviations: white arrowheads indicate neurons of the cerebral ganglion; blue arrowheads indicate immunopositive cells in the palps; white arrows indicate newly grown fibers; white dashed line marks the cut line; yellow dotted line marks the border of the growing food grove; *br*—brain; *cc*—circumesophageal connectives; *cco*—cerebral commissure; *dn*—dorsal longitudinal nerve; *fg*—food grove; *ln*—lateral longitudinal nerve; *no*—nuchal organ; *pbn*—peripheral nerves in the body wall of the regenerate; *pln*—palp nerve; *sg*—segmental ganglion; *sn*—segmental nerve; *sog*—subesophageal ganglion; and *vnc*—ventral nerve cord. The head is to the left and top; 5-HT-positive elements—green; acetylated α-tubulin-like elements—magenta. Double-sided a-p arrows indicate the direction of the anteroposterior axis.

**Figure 10 biology-12-01412-f010:**
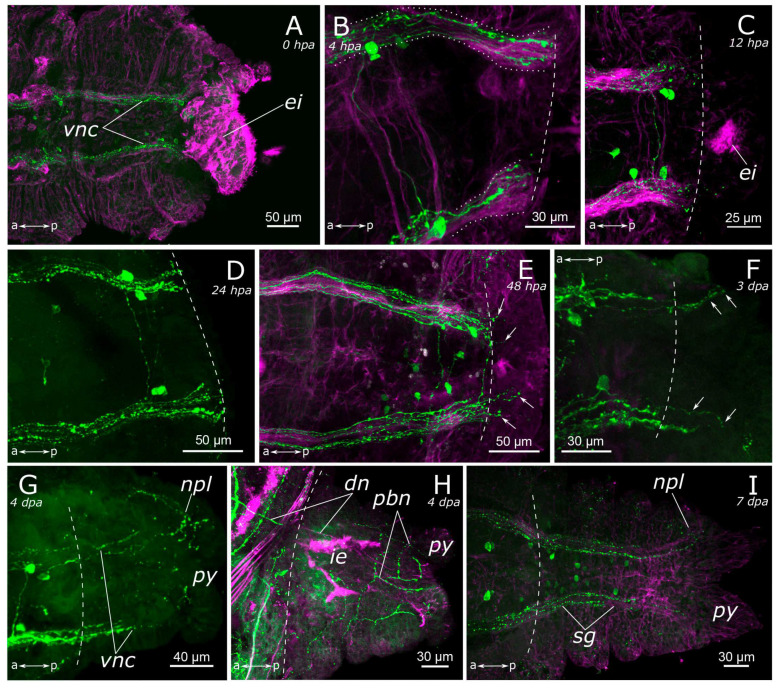
Posterior regeneration of the 5-HT-positive elements of the *Pygospio elegans* nervous system. (**A**) 0 hpa, ventral view; (**B**) 4 hpa, ventral view; (**C**) 12 hpa, ventral view; (**D**) 24 hpa, ventral view; (**E**) 48 hpa, ventral view; (**F**) 3 dpa, ventral view; (**G**) 4 dpa, ventral view; (**H**) 4 dpa, dorsal view; (**I**) 7 dpa, ventral view. Abbreviations: white arrows indicate newly grown fibers; white dashed line marks the cut line; white dotted line marks the ventral nerve cords and its frayed ends; *dn*—dorsal longitudinal nerve; *ei*—everted intestine; *ie*—intestinal epithelium; *npl*—nerves of pygidial lobes; *pbn*—peripheral nerves in the body wall of the regenerate; *py*—pygidium; *sg*—segmental ganglion; and *vnc*—ventral nerve cord. “Tail” is to the right; 5-HT-positive elements—green; and acetylated α-tubulin-like elements—magenta. Double-sided a-p arrows indicate the direction of the anteroposterior axis.

**Figure 11 biology-12-01412-f011:**
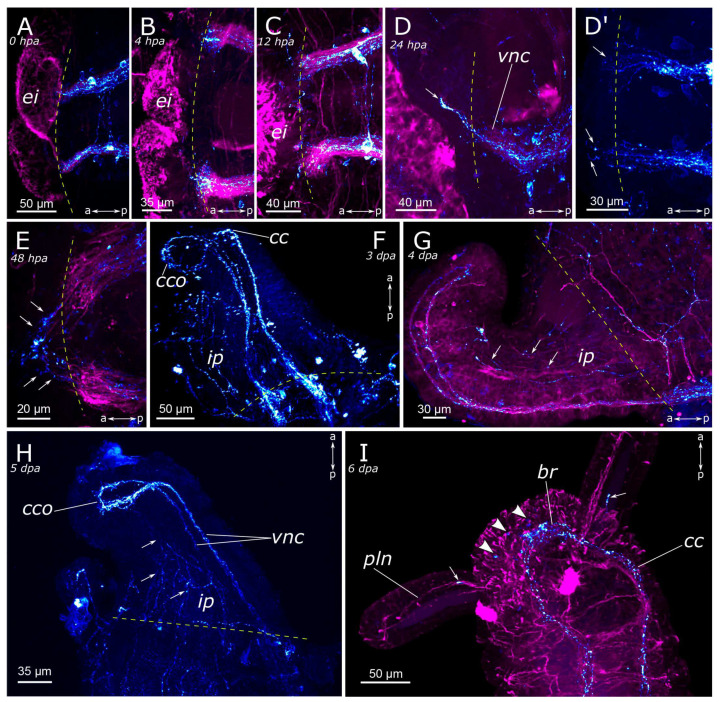
Early and middle stages of anterior regeneration of FMRFamide-positive elements of the *Pygospio elegans* nervous system. (**A**) 0 hpa, ventral view; (**B**) 4 hpa, ventral view; (**C**) 12 hpa, ventral view; (**D**) 24 hpa, lateral view; (**D’**) 24 hpa ventral view; (**E**) 48 hpa, ventral view; (**F**) 3 dpa, ventral view; (**G**) 4 dpa, lateral view; (**H**) 5 dpa, lateral view; (**I**) 6 dpa, ventral view. Abbreviations: white arrows indicate newly grown fibers; white arrowheads indicate neurons; yellow dashed line marks the cut line; *br*—brain; *cc*—circumesophageal connectives; *cco*—cerebral commissure; *ei*—everted intestine; *ip*—intestinal plexus; *pln*—palp nerve; and *vnc*—ventral nerve cord. The head is to the left and top; FMRFamide-positive elements—cyan, and acetylated α-tubulin-like elements—magenta. Double-sided a-p arrows indicate the direction of the anteroposterior axis.

**Figure 12 biology-12-01412-f012:**
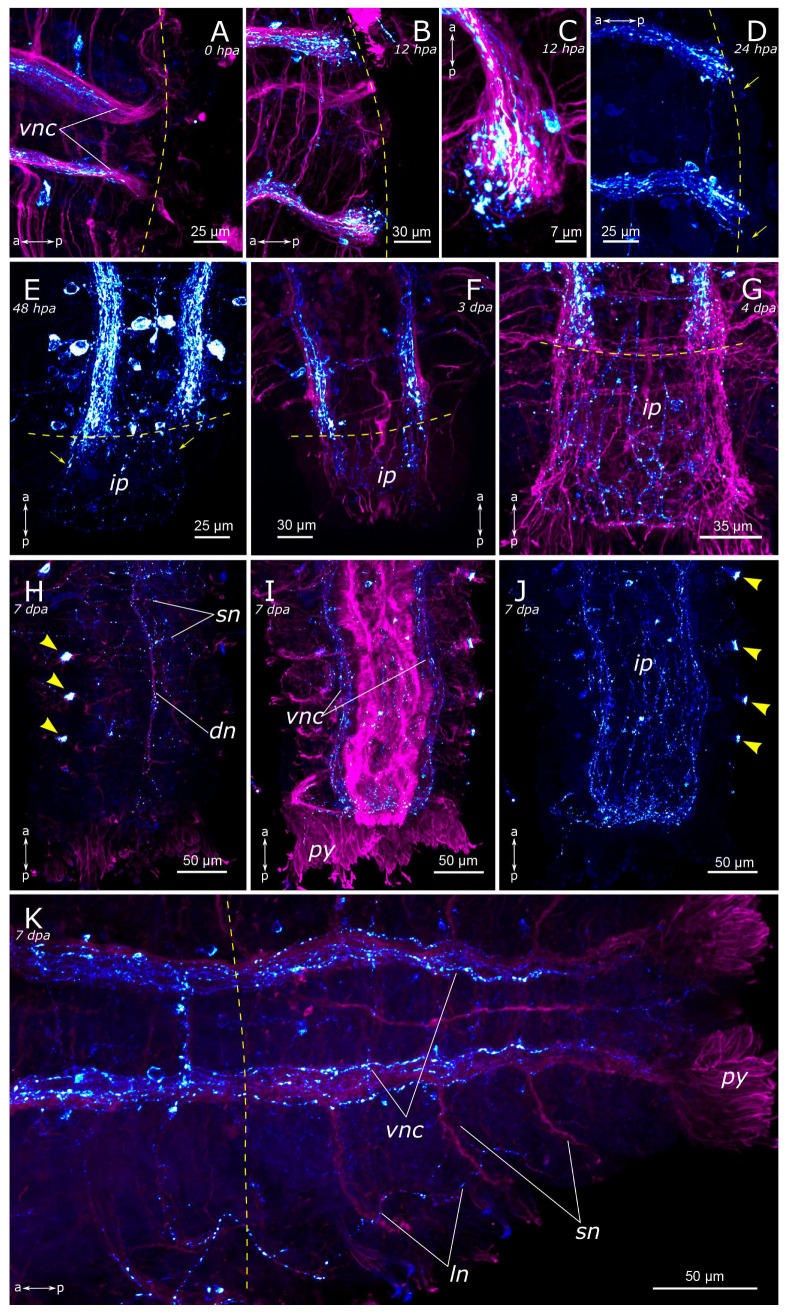
Posterior regeneration of FMRFamide-positive elements of the *Pygospio elegans* nervous system. (**A**) 0 hpa, ventral view. (**B**) 12 hpa, ventral view. (**C**) 12 hpa, frayed ends of the ventral nerve cord. (**D**) 24 hpa, ventral view. (**E**) 48 hpa, ventral view. (**F**) 3 dpa, ventral view. (**G**) 4 dpa, ventral view. (**H**–**J**) 7 dpa, partial z-projections through the growing tail: (**H**) 7 dpa, dorsal view; (**I**) 7 dpa, ventral view; (**J**) 7 dpa, gut innervation. (**K**) 9 dpa, ventral view. Abbreviations: yellow arrows—newly grown fibers; yellow arrowheads—neurons; yellow dashed line marks the cut line; *dn*—dorsal longitudinal nerve; *ip*—intestinal plexus; *ln*—lateral longitudinal nerve; *py*—pygidium; *sn*—segmental nerve; *vnc*—ventral nerve cord; “Tail” is to the bottom and right; FMRFamide-positive elements—cyan; and acetylated α-tubulin-like elements—magenta. Double-sided a-p arrows indicate the direction of the anteroposterior axis.

**Figure 13 biology-12-01412-f013:**
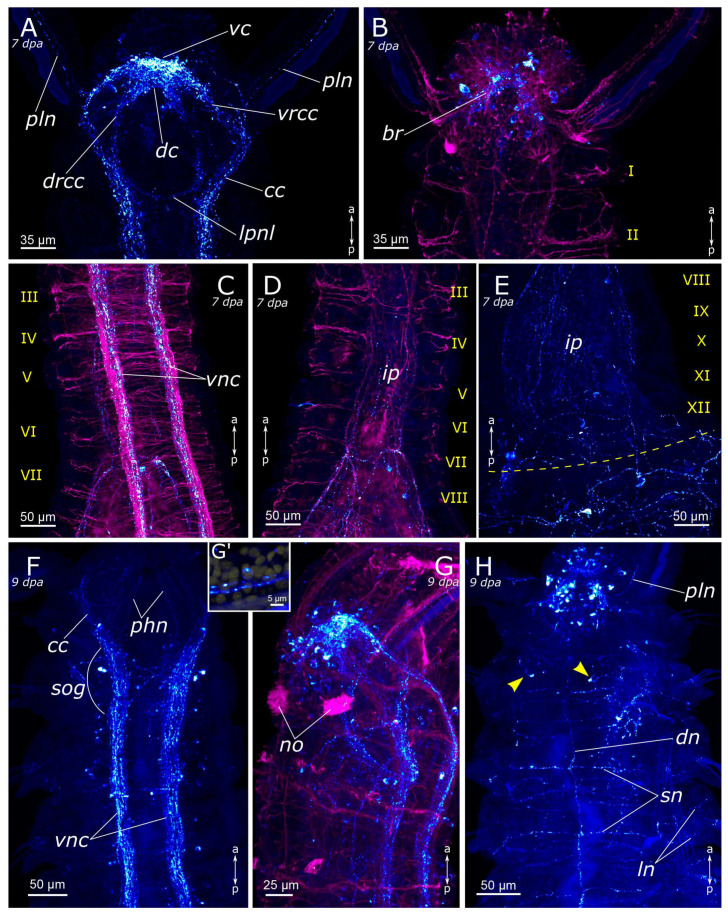
Late stages of anterior regeneration of FMRFamide-positive elements of the *Pygospio elegans* nervous system. (**A**) 7 dpa, ventral view; (**B**) 7 dpa, dorsal view; (**C**) 7 dpa, growing head from 3rd to 7th segments, ventral view; (**D**) 7 dpa, growing head from 3rd to 8th segments; (**E**) 7 dpa, growing head from 8th to 12th segments; (**F**) 9 dpa, ventral view; (**G**) 9 dpa, lateral view; (**G’**) 9 dpa, FMRFamide-positive cell in the palp near the palp nerve; (**H**) 9 dpa, dorsal view. Abbreviations: yellow arrowheads indicate receptor cells; yellow dashed line marks the cut line; roman numerals mark the segments; *br*—brain; *cc*—circumesophageal connectives; *dc*—dorsal cerebral commissure; *dn*—dorsal longitudinal nerve; *drcc*—dorsal roots of the circumesophageal connectives; *ip*—intestinal plexus; *ln*—lateral longitudinal nerve; *lpnl*—lateral pharyngeal nerve loop; *no*—nuchal organ; *phn*—pharyngeal nerves; *pln*—palp nerve; *sn*—segmental nerve; *sog*—subesophageal ganglion; *vc*—ventral cerebral commissure; *vnc*—ventral nerve cord; and *vrcc*—ventral roots of the circumesophageal connectives. The head is to the top; FMRFamide-positive elements—cyan; acetylated α-tubulin-like elements—magenta; and DAPI—yellow. Double-sided a-p arrows indicate the direction of the anteroposterior axis.

**Figure 14 biology-12-01412-f014:**
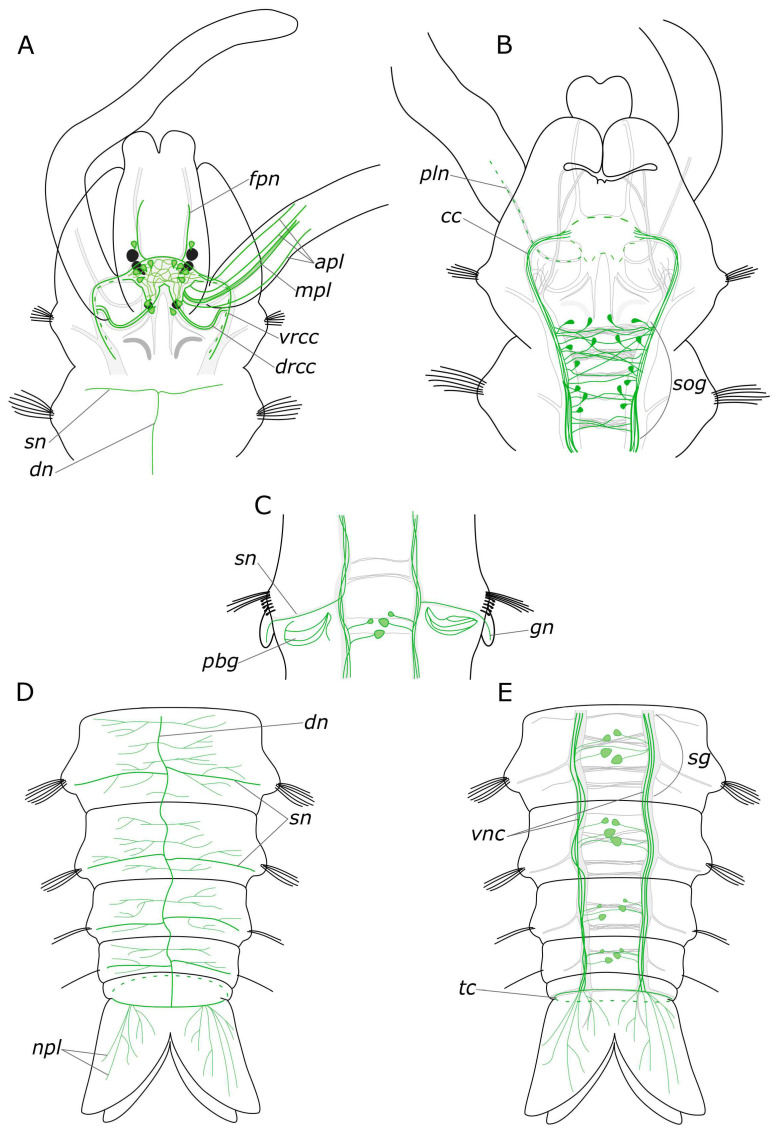
Schematic representation of 5-HT-positive elements (green) in the *Pygospio elegans* nervous system. The main nerve tracts of the central nervous system are outlined in grey. (**A**) Anterior part, dorsal view. (**B**) Anterior part, ventral view. (**C**) The abdominal segment with gills, ventral view. (**D**) Posterior part, dorsal view. (**E**) Posterior part, ventral view. Abbreviations: *apl*—additional palp nerve, *cc*—circumesophageal connectives; *dn*—dorsal longitudinal nerve; *drcc*—dorsal roots of the circumesophageal connectives; *fpn*—frontal prostomial nerve; *gn*—gill nerve; *mpl*—main palp nerve; *npl*—nerves of pygidial lobes; *pln*—palp nerve; *sg*—segmental ganglion; *sog*—subesophageal ganglion; *sn*—segmental nerve; *tc*—terminal commissure; *vnc*—ventral nerve cord; and *vrcc*—ventral roots of the circumesophageal connectives.

**Figure 15 biology-12-01412-f015:**
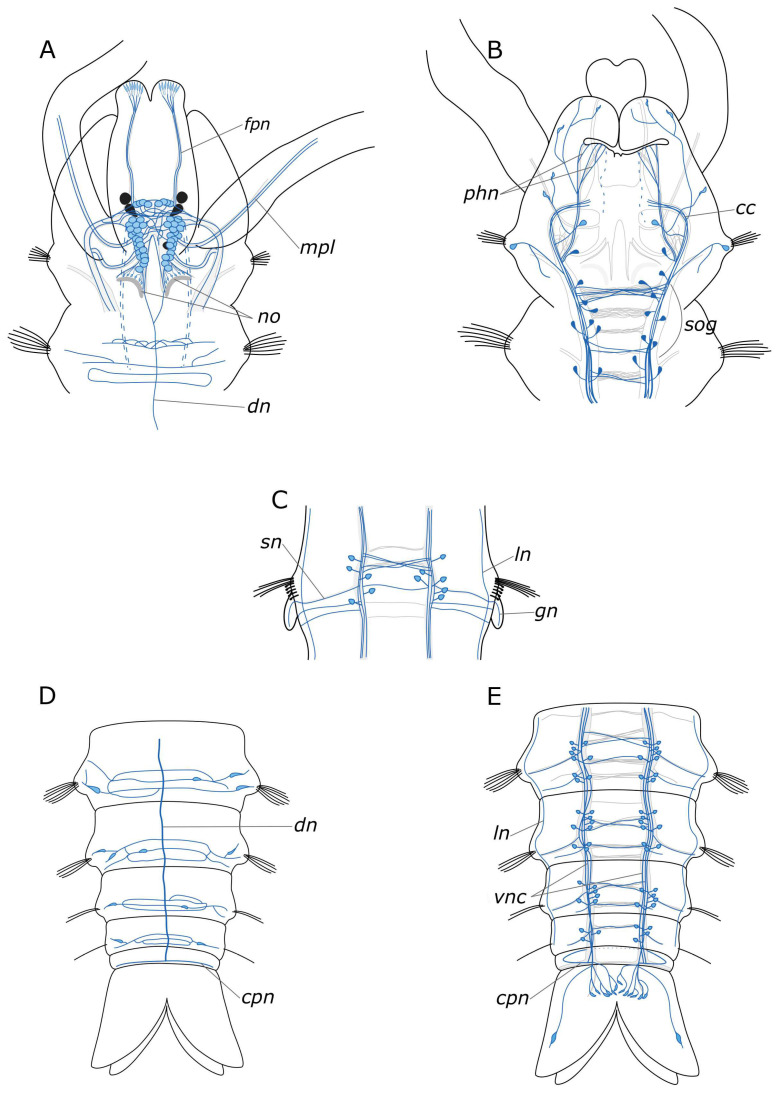
Schematic representation of FMRFamide-positive elements (blue) in the *Pygospio elegans* nervous system. The main nerve tracts of the central nervous system are outlined in grey. (**A**) Anterior part, dorsal view. (**B**) Anterior part, ventral view. (**C**) An abdominal segment with gills, ventral view. (**D**) Posterior part, dorsal view. (**E**) Posterior part, ventral view. Abbreviations: *cc*—circumesophageal connectives; *cpn*—circumpygidial nerve; *dn*—dorsal longitudinal nerve; *fpn*—frontal prostomial nerve; *gn*—gill nerve; *ln*—lateral longitudinal nerve; *mpl*—main palp nerve; *no*—nuchal organ; *phn*—pharyngeal nerves; *sog*—subesophageal ganglion; *sn*—segmental nerve; and *vnc*—ventral nerve cord.

**Figure 16 biology-12-01412-f016:**
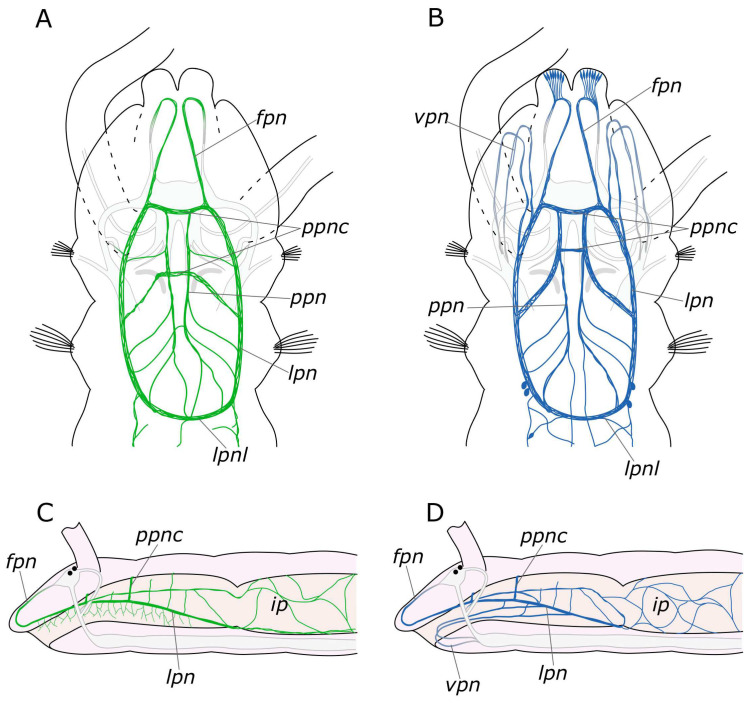
Schematic representation of the stomatogastric nervous system of the *Pygospio elegans*. (**A**,**C**) 5-HT-positive elements (outlined in green). (**B**,**D**) FMRFamide-positive elements (outlined in blue). The main nerve tracts of the central nervous system are outlined in grey. Abbreviations: *fpn*—frontal prostomial nerve; *ip*—intestinal plexus; *lpn*—lateral pharyngeal nerves; *lpnl—*lateral pharyngeal nerve loop; *ppn*—paramedian pharyngeal nerves; *ppnc*—paramedian pharyngeal nerves commissure; and *vpn*—ventral pharyngeal nerves.

**Figure 17 biology-12-01412-f017:**
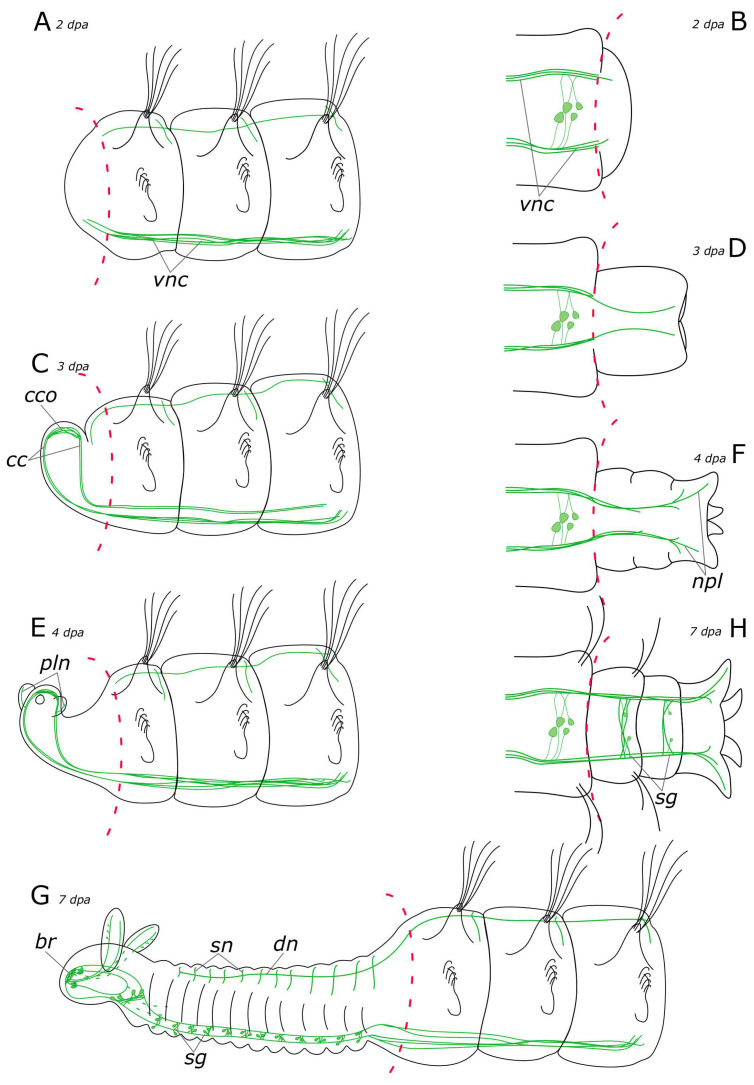
Schematic representation of 5-HT-positive elements (green) in the *Pygospio elegans* nervous system regeneration. (**A**) Anterior regeneration 2 dpa. (**B**) Posterior regeneration 2 dpa. (**C**) Anterior regeneration 3 dpa. (**D**) Posterior regeneration 3 dpa. (**E**) Anterior regeneration 4 dpa. (**F**) Posterior regeneration 4 dpa. (**G**) Anterior regeneration 7 dpa. (**H**) Posterior regeneration 7 dpa. Abbreviations: *br*—brain; *cc*—circumesophageal connectives; *cco*—cerebral commissure; *dn*—dorsal longitudinal nerve; *npl*—nerves of pygidial lobes; *pln*—palp nerve; *sg*—segmental ganglion; *sn*—segmental nerve; and *vnc*—ventral nerve cord.

**Figure 18 biology-12-01412-f018:**
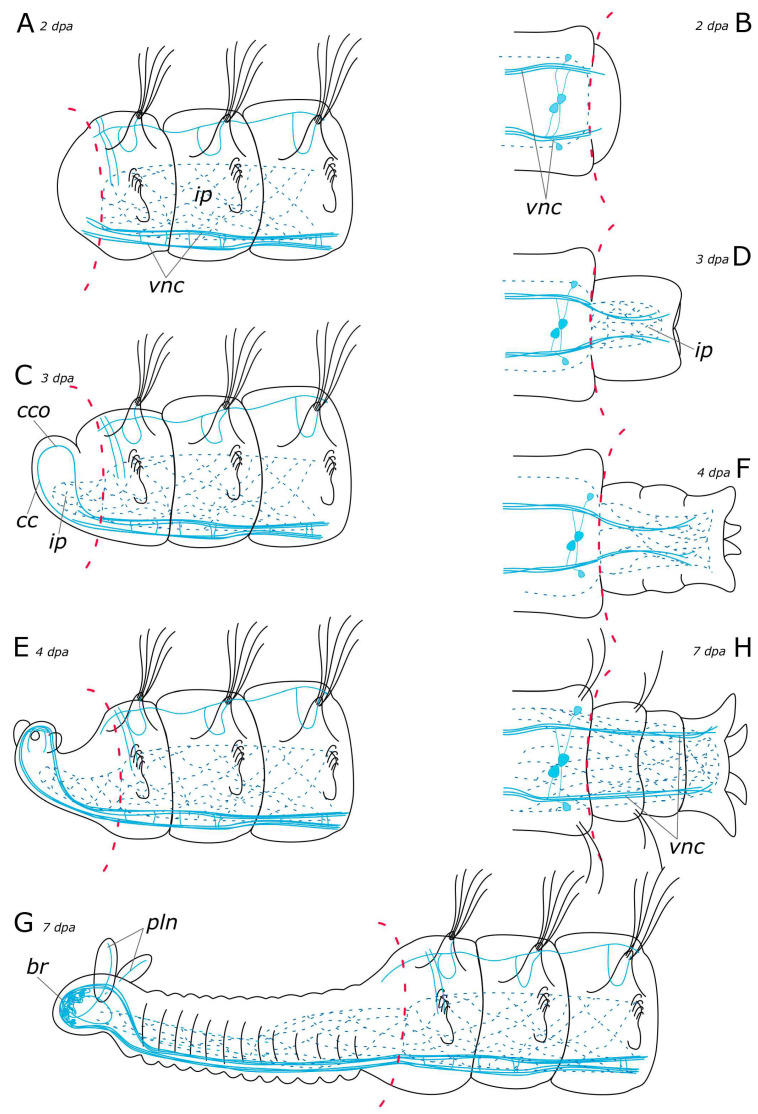
Schematic representation of FMRFamide-positive elements (blue) in the *Pygospio elegans* nervous system regeneration. (**A**) Anterior regeneration 2 dpa. (**B**) Posterior regeneration 2 dpa. (**C**) Anterior regeneration 3 dpa. (**D**) Posterior regeneration 3 dpa. (**E**) Anterior regeneration 4 dpa. (**F**) Posterior regeneration 4 dpa. (**G**) Anterior regeneration 7 dpa. (**H**) Posterior regeneration 7 dpa. Abbreviations: *br*—brain; *cc*—circumesophageal connectives; *cco*—cerebral commissure; *ip*—intestinal nerve plexus; *pln*—palp nerve; and *vnc*—ventral nerve cord.

**Table 1 biology-12-01412-t001:** Distribution of different neurotransmitters in neuroregeneration of *Pygospio elegans.* Abbreviations: AR—anterior regeneration; *cc*—circumesophageal commissure; *dn*—dorsal longitudinal nerve; *ln*—lateral longitudinal nerves; *npl*—nerves of pygidial lobes; NS—nervous system; *pbn*—peripheral nerves in the body wall of the regenerate; *pln*—palp nerves; PR—posterior regeneration; *prc*—palp receptor cells; SG—segmental ganglion; *sn*—segmental nerves; and VNC—ventral nerve cord.

	5-HT-pos. NS		FMRFamide-pos. NS		GABA-pos. NS (after [43])		HA-pos. NS (after [43])		CA-pos. NS (after [10])	
	AR	PR	AR	PR	AR	PR	AR	PR	AR	PR
1 dpa, wound healing	Single fibers of VNC	–	Single fibers of VNC	–	–	–	–	–	–	–
2 dpa, blastema	Fibers of VNC	Fibers of VNC	Fibers of VNC	Fibers of VNC	–	–	–	–	Single fibers of VNC	–
3 dpa, primordia	Neuropil, *cc*, VNC, *pbn*	VNC	Neuropil, *cc*, VNC, intestinal plexus	VNC, intestinal plexus	Single fibers of VNC	Dorsal longitudinal nerve	Single fibers of VNC	Single fibers of VNC	Sensory cells and fibers, fibers of VNC	Fibers of VNC
4–6 dpa, segmentation	(4–6 dpa) Neuropil, *cc*, VNC, *pln*, *dn*, *sn*	(4–6 dpa) VNC, *npl*, *pbn*	(4–6 dpa) Neuropil, *cc*, VNC, brain neurons, intestinal plexus, *pln*	(4–6 dpa) VNC, intestinal plexus	(4 dpa) VNC, brain neurons	(4 dpa) VNC	(4 dpa) VNC, intestinal nerve	(4 dpa) VNC	(4 dpa) Sensory cells, dorsal nerve	(4 dpa) VNC, nerve elements of the pygidium
7–9 dpa, differentiation	(7–9 dpa) Cerebral ganglion, SG, *pln*, *prc*, *dn*, *sn*, *ln*	(7–9 dpa) SG, *npl*	(7–9 dpa) Cerebral ganglion, VNC, *pln*, *prc*, *dn*, *sn*, *ln*	(7–9 dpa) SG, *ln*, *sn*	(7 dpa) Cerebral ganglion, SG, *pln*, *prc*	(7 dpa) SG	(7 dpa) Cerebral ganglion, VNC, *pln*	(7 dpa) VNC	(7 dpa) Cerebral ganglion, VNC, *pln*	(7 dpa) VNC, nerve elements of the pygidium, *sn*

## Data Availability

The datasets analyzed during this study are available from the corresponding author on reasonable request. All the necessary data are presented in the paper.

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
