# Peer review of "Mass Start or Time Trial? Structure of the Nervous System and Neuroregeneration in Pygospio elegans (Spionidae, Annelida)"

_biology, 2023, doi:10.3390/biology12111412_

Round 1

Reviewer 1 Report

Comments and Suggestions for Authors

In the manuscript “biology-2676339” entitled “Mass start or time trail? Structure of the nervous system and neuroregeneration in Pygospio elegans (Spionidae, Annelida)” authors descriptively studied the structure of Pygospio elegans nervous system and its regeneration through multiple time points following amputation. They have analysed 5-HT- and FMRFamide-positive nerve elements by applying antibodies. Authors have discussed their results through comparison with previous findings regarding other types of neurons in P. elegans. Generally, the study is well designed and presented. The content is delivered using an appropriate level of language. The figures are well presented, however, the quality could be improved. Altogether, the descriptive nature of the study, I believe this manuscript is suitable for publishing in journal Biology.

Minor comments: 

1- A consistency of using P. elegans throughout the manuscript is required. For example line 463 authors use the full name.

2- Full name of D. quadrilobata needs to be mentioned in the text body (line 527).

Additional Comments:

1. What is the main question addressed by the research?

I addressed this question in my comments. They have studied neuronal

structure of two types of neurons (serotonin and FMRFamide) in Pygospio

elegans.

The nature of their study is descriptive, hence, the way of description is

important.

2. Do you consider the topic original or relevant in the field? Does it

address a specific gap in the field?

They describe the structure and regeneration of neurons of an animal model.

To the best of my knowledge it is original, and has the chance to be

relevant to the journal's scope.

3. What does it add to the subject area compared with other published

material?

It describes two types of neurons (serotonin and FMRFamide) in Pygospio

elegans, which is new information.

4. What specific improvements should the authors consider regarding the

methodology? What further controls should be considered?

It has been answered already.

5. Are the conclusions consistent with the evidence and arguments presented

and do they address the main question posed?

Yes.

6. Are the references appropriate?

Yes.

7. Please include any additional comments on the tables and figures.

They have been already added as minor comments.

Author Response

We thank Reviewer very much for the review and valuable feedback on our manuscript. We sincerely appreciate positive assessment of our work, and we have made the necessary improvements to enhance the figure quality.

Reviewer: The figures are well presented, however, the quality could be improved.

Answer: We made every effort to ensure that the quality of our illustrations is the best. Nevertheless, we are ready to improve the quality of our illustrations by following specific suggestions.

Reviewer: A consistency of using P. elegans throughout the manuscript is required. For example line 463 authors use the full name.

Answer: Corrected

2- Full name of D. quadrilobata needs to be mentioned in the text body (line 527).

Answer: Corrected

Reviewer 2 Report

Comments and Suggestions for Authors

Sunkina and colleagues have taken an immunofluorescence approach to investigate the nervous system of Pygospio elegans. While this has been performed in other annelids using similar approaches and in P. elegans using different antibodies, this study adds new information regarding the location of neurons containing serotonin and FMRFamides. The authors then detail the locations and changes in staining patterns during a weeklong regeneration period after amputation. This study includes high quality data and diagrams to explain the locations of neurons containing serotonin or FMRFamides. However, there are concerns about the rigor and reproducibility of the results.

Major Concerns:

1. The authors state that at least 10 animals were used for each antibody at each regeneration stage, but the sample size of uninjured animals is not noted.

2. While the images and descriptions of the data provide a thorough explanation of the observed staining patterns, there is no analysis of how reproducible these reported results are. Do these patterns appear the same in all animals investigated? Are their deviations between animals? Scoring or other modes of quantification of the staining patterns before and after amputation would increase the rigor of the study and provide a better assessment of the reproducibility of the authors' observations.

3. The authors document different stages of regeneration but do not include controls for the amputation or at the least do not display these controls for comparison with the stages of regeneration. Additionally, brightfield images could be useful for assessing the site of amputation and tissue damage.

Minor Concerns:

1. An anatomical diagram prior to or in Figure 1 could aid readers understand and appreciate the locations of the various staining patterns. Figures 13-17 were very helpful for understanding the locations of neurons and processes.

2. Figure 2 has dashed lines outside the left edge of panel C.

3. Figure 7 has an outlined "B" above the left-hand corner of panel B.

Comments on the Quality of English Language

There were a few typos or grammatical errors and some instances of odd word choices in the text. These were only minor concerns since the majority of the study was very well-written.

Author Response

We appreciate Reviewer thoughtful review and constructive feedback on our article. Reviewer’s concerns are valuable, and we are grateful for the opportunity to enhance the quality and rigor of our study. We addressed the concerns raised and make the necessary improvements, including adding of an extra Figure with general morphology, as suggested. Reviewer input has undoubtedly contributed to the overall quality of our work.

Reviewer: The authors state that at least 10 animals were used for each antibody at each regeneration stage, but the sample size of uninjured animals is not noted.

Answer: To describe the general structure of the nervous system, 20 adult animals without damage or signs of regeneration were stained. We really did not indicate this point in the materials and methods. This information has been added to the manuscript.

Reviewer: While the images and descriptions of the data provide a thorough explanation of the observed staining patterns, there is no analysis of how reproducible these reported results are. Do these patterns appear the same in all animals investigated? Are their deviations between animals? Scoring or other modes of quantification of the staining patterns before and after amputation would increase the rigor of the study and provide a better assessment of the reproducibility of the authors' observations.

Answer: Antibody staining might exhibit slight variation between samples. Elements of the central nervous system tend to display consistent and minimal variability in staining. Whereas when identifying peripheral nervous system elements, some variability between samples may be observed. Moreover, we found no differences in the structure and staining of the nervous system between the intact part of the regenerated worm and uninjured animals. Elements of the nervous system that were unique to a single individual were not included in the results. According to our experience with various antibodies in P. elegans, the clearest picture was obtained by antibodies against serotonin, FMRFamide and acetylated α-tubulin. In contrast, antibodies against histamine, GABA and octopamine produced more variable outcomes (Starunova et al., 2022, Starunova et al., 2023, in press). Our studies on both the general morphology and morphology of the nervous system of P. elegans are comparable to each other. The nervous system topology identified with antibodies aligns with that revealed by other methods (Starunov et al., 2020, Barmasova et al., 2022). We added the information on the reproducibility of results and comparison of staining in uninjured animals and the intact part of regenerating worms to the materials and methods section.

Reviewer: The authors document different stages of regeneration but do not include controls for the amputation or at the least do not display these controls for comparison with the stages of regeneration. Additionally, brightfield images could be useful for assessing the site of amputation and tissue damage.

Answer: We did not affect the animals to any experimental treatments beyond the amputation process. To assess nervous system regeneration, we used uninjured animals as controls, and we have provided a comprehensive description of their neuroanatomy. Notably, we observed no differences in the nervous system structure between uninjured animals and the intact part of regenerating worms.

We standardized the methodology for regeneration experiments to ensure comparability across various studies. All regeneration experiments maintained consistent conditions, including temperature, water salinity, illumination, keeping animals in individual dishes without sand and feeding. Worms were selected for all experiments of the same size and number of segments, without visible damage or signs of regeneration. Daily visual monitoring of regeneration progress was performed, with non-compliant individuals promptly removed. In our previous study (Starunov et al., 2020), we provided detailed description on the overall regeneration process, wound healing (SEM), muscle and nervous system restoration in P. elegans. Our results on the stages of regeneration, correspond to those described previously in the literature, though adjusted for different laboratory conditions (Lindsay et al., 2007; Lindsay et al., 2008). According to the literature, regeneration after operation and regeneration as a result of fragmentation in nature do not differ in the main stages in P. elegans (Gibson, Harvey, 2000).

Reviewer: An anatomical diagram prior to or in Figure 1 could aid readers understand and appreciate the locations of the various staining patterns. Figures 13-17 were very helpful for understanding the locations of neurons and processes.

Answer: We have added one more figure (Figure 1 in revised manuscript) with the general structure of the worm and regeneration experiment to the Materials and Methods section.

Reviewer: Figure 2 has dashed lines outside the left edge of panel C.

Answer: Corrected

Reviewer: Figure 7 has an outlined "B" above the left-hand corner of panel B.

Answer: Corrected

Reviewer 3 Report

Comments and Suggestions for Authors

Sunkina et al. described the regeneration process of Pygospio elegans following amputation. They focused on the serotonergic and FMRFamidergic nervous systems. I have some comments mainly on the description in Material & Methods and the way of displaying the images in this manuscript.

1. The age of the worms is not described. This may be because they were caught in wild. How do the authors control the conditions on the age of them? In general, the age of animals significantly affects the speed of regeneration.

2. line 106-112: Why do the authors omit the use of surfactant? It would be difficult for antibodies to penetrate into cells without the use of TritonX-100 or Tween-20 in immunohistochemistry.

3. According to the description in Material & Methods, the authors used laser scanning microscopy to obtain fluorescence images. How were the tissues cleared? Otherwise, were they sectioned in cryostat?

3. How was the specificity of anti-5HT ensured? I understand that the antibody was commercially obtained, and it is expected to react with 5-HT in the tissue. But the binding to other molecules cannot be denied in this experimental condition (i.e. animal and way of fixation). Non-specific binding of secondary antibody was also not excluded yet. Further grounds are needed to demonstrate the validity of the use of this antibody in this animal and fixation method. In general, coincident expression of serotonin transporter mRNA (in situ hybridization) strengthens the data obtained with immunohistochemistry. Otherwise, pre-adsorption with antigen would help in immunohistochemistry.

4. line 119, 120: Insert the word “IgG” just following Anti-Rabbit and Anti-Mouse.

5. Axes, such as anterior-posterior, dorsal-ventral, are necessary for all Figures.

Comments on the Quality of English Language

I found no problem.

Author Response

We are grateful to the reviewer for the thoughtful attention to our manuscript and valuable feedback provided. We have taken Reviewer comments into consideration and made the necessary revisions to address the points were raised. We believe Reviewer insights contributed to the overall improvement of our work.

Reviewer: The age of the worms is not described. This may be because they were caught in wild. How do the authors control the conditions on the age of them? In general, the age of animals significantly affects the speed of regeneration.

Answer: In our experiments, we used criteria based on the segment number rather than age-related ones. Information on the number of segments in worms in experiments has been added to materials and methods section. Therefore, we've selected medium-sized worms (approximately 40 segments) to ensure the exclusion of juveniles and excessively large or aged individuals. We collected adult worms from intertidal zone and subsequently maintain them in laboratory culture for approximately a year. Available literature suggests that P. elegans typically lives for 2-3 years (Armatage, 1977). Determining their precise age is challenging since P. elegans exhibits spontaneous fragmentation followed by regeneration, in both natural and laboratory conditions. Therefore, there is no way to use age parameters for this species.

Reviewer: line 106-112: Why do the authors omit the use of surfactant? It would be difficult for antibodies to penetrate into cells without the use of TritonX-100 or Tween-20 in immunohistochemistry.

Answer: Thank you for this comment, we just forgot to write about it. For almost all labeling procedures we used 0.1M PBS with 0.1% Triton X-100 (PBT). We have added this information to materials and methods section.

Reviewer: According to the description in Material & Methods, the authors used laser scanning microscopy to obtain fluorescence images. How were the tissues cleared? Otherwise, were they sectioned in cryostat?

Answer: P. elegans are small, lightly pigmented worms with a non-dense cuticle, making them well-suited for labeling and visualization as whole-mounts. The penetration of the antibodies as well as the other dyes is good enough and makes it possible to analyze the specimens at their whole depth. Our labeling process does not require sectioning prior to antibody application. We employ Mowiol as the mounting medium. This medium effectively cleared our specimens within a few days.

Reviewer: How was the specificity of anti-5HT ensured? I understand that the antibody was commercially obtained, and it is expected to react with 5-HT in the tissue. But the binding to other molecules cannot be denied in this experimental condition (i.e. animal and way of fixation). Non-specific binding of secondary antibody was also not excluded yet. Further grounds are needed to demonstrate the validity of the use of this antibody in this animal and fixation method. In general, coincident expression of serotonin transporter mRNA (in situ hybridization) strengthens the data obtained with immunohistochemistry. Otherwise, pre-adsorption with antigen would help in immunohistochemistry.

Answer: Thank you for this comment. We performed a series of control experiments to ensure the specificity of our antibody labeling. The application of secondary antibodies without previous incubation with any primary antibody did not reveal any labeling. We used these secondary antibodies with a broad range of primary ones and in all cases we did not find any non-specific labeling. In addition, we tested another pair of secondary antibodies (CF 488 donkey anti-mouse IgG and CF633 donkey anti-rabbit IgG) which resulted the same labeling results. We also made pre-adsorption experiments with 50 µg/ml Serotonin hydrochloride (Sigma, H9523) and 40 µg/ml Serotonin BSA Conjugate Control (Immunostar, 20081; the control provided and recommended by antibody manufacturer). In both cases application of the control led to the completely elimination of labeling in the whole range of the used dilutions (1:2000, 1:4000; 1:10000, 1:20000, and 1:40000). In the series of parallel positive controls (without serotonin hydrochloride or serotonin-BSA) the specific labeling was detected in the whole range of dilutions, though the signal of 1:10000 – 1:40000 dilutions was gradually weaker. We added the requested information to the manuscript text.

Reviewer: line 119, 120: Insert the word “IgG” just following Anti-Rabbit and Anti-Mouse.

Answer: Corrected

Reviewer: Axes, such as anterior-posterior, dorsal-ventral, are necessary for all Figures.

Answer: Corrected

Round 2

Reviewer 2 Report

Comments and Suggestions for Authors

The author's revisions appropriately attend to and corrected all the reviewer's concerns.

Reviewer 3 Report

Comments and Suggestions for Authors

The Methods section of the ms was substantually improved according to my suggestions. Addition of a new figure is appreciated.